# Context-dependent requirement of G protein coupling for Latrophilin-2 in target selection of hippocampal axons

Daniel T Pederick[1†], Nicole A Perry-Hauser[2,3†], Huyan Meng[4], Zhigang He[4], Jonathan A Javitch[2,3*], Liqun Luo[1*]

[1]Department of Biology, Howard Hughes Medical Institute, Stanford University, Stanford, United States; [2]Departments of Psychiatry and Molecular Pharmacology and Therapeutics, Columbia University Vagelos College of Physicians and Surgeons, New York, United States; [3]Division of Molecular Therapeutics, New York State Psychiatric Institute, New York, United States; [4]F.M. Kirby Neurobiology Center, Department of Neurology, Boston Children's Hospital, Harvard Medical School, Boston, United States

*For correspondence:
jaj2@cumc.columbia.edu (JAJ);
lluo@stanford.edu (LL)

†These authors contributed equally to this work

Competing interest: The authors declare that no competing interests exist.

**Abstract** The formation of neural circuits requires extensive interactions of cell-surface proteins to guide axons to their correct target neurons. *Trans*-cellular interactions of the adhesion G protein-coupled receptor latrophilin-2 (Lphn2) with its partner teneurin-3 instruct the precise assembly of hippocampal networks by reciprocal repulsion. Lphn2 acts as a repulsive receptor in distal CA1 neurons to direct their axons to the proximal subiculum, and as a repulsive ligand in the proximal subiculum to direct proximal CA1 axons to the distal subiculum. It remains unclear if Lphn2-mediated intracellular signaling is required for its role in either context. Here, we show that Lphn2 couples to $G\alpha_{12/13}$ in heterologous cells; this coupling is increased by constitutive exposure of the tethered agonist. Specific mutations of Lphn2's tethered agonist region disrupt its G protein coupling and autoproteolytic cleavage, whereas mutating the autoproteolytic cleavage site alone prevents cleavage but preserves a functional tethered agonist. Using an in vivo misexpression assay, we demonstrate that wild-type Lphn2 misdirects proximal CA1 axons to the proximal subiculum and that Lphn2 tethered agonist activity is required for its role as a repulsive receptor in axons. By contrast, neither tethered agonist activity nor autoproteolysis were necessary for Lphn2's role as a repulsive ligand in the subiculum target neurons. Thus, tethered agonist activity is required for Lphn2-mediated neural circuit assembly in a context-dependent manner.

## Editor's evaluation

This is an intriguing study investigating the molecular mechanisms of neural circuit developmental organization. Using a defined hippocampal circuit, the authors find that ectopic expression of an adhesion G protein receptor leads to axon mistargeting. This work defines new mechanisms of axon target specificity.

## Introduction

Latrophilins (Lphn1–3) are highly expressed in the brain and were originally identified as responders to α-latrotoxin, a neurotoxin from black widow spider venom that causes the profound release of neurotransmitters from nerve terminals (*Davletov et al., 1996*). They belong to the family of adhesion G protein-coupled receptors (aGPCRs), capable of eliciting intracellular effects through coupling with

**eLife digest** The complex brain circuits that allow animals to sense and interact with their environment start to form early during development. Throughout this period, neurons extend fiber-like projections to establish precise wiring patterns. Various types of proteins at the surface of both incoming fibers and target cells ensure that only the right partners will connect together.

Latrophilin-2, for example, is a neuronal surface protein essential for the formation of accurate connections in the hippocampus, a brain region important for memory. Studded through the membrane of certain neurons, it acts as a signal-sending ligand to direct incoming fibers, with neurons that carry Latrophilin-2 repelling projections from cells that display certain protein partners.

At the same time, Latrophilin-2 also allows neurons to receive chemical signals by working with intracellular signaling proteins known as G proteins, which help to relay information between cells. It remained unclear how this role as a signalling receptor participates in the wiring of the hippocampus during development.

To explore this question, Pederick, Perry-Hauser et al. examined the impact of Latrophilin-2 on the connection patterns of mouse hippocampal neurons that do not normally carry this protein. Introducing Latrophilin-2 into these 'proximal CA1 cells' misdirected them away from their usual partners – unless Latrophilin-2 was altered so that it could not interact with G proteins. In contrast, forcing the connecting partners of CA1 cells to display normal or altered versions of Latrophilin-2 did not interfere with the protein acting as a repulsive ligand. Taken together, these results suggest that the ability of Latrophilin-2 to signal through G proteins is important for neurons that are attempting to project their fibers onto other cells, but not important when Latrophilin-2 acts in targets to direct incoming fibers from other neurons.

These results show that a single protein can shape neural circuits by acting both as a signal-receiving receptor and a signal-sending ligand depending on the context. In the future, Pederick, Perry-Hauser et al. hope that their findings will shed new light on how the wiring of the brain is disrupted in neurodevelopmental disorders.

heterotrimeric G proteins (*Lelianova et al., 1997*). Additionally, as cell adhesion molecules, latrophilins interact via their N-terminal extracellular domain with four different families of interacting partners including neurexins (*Boucard et al., 2012*), teneurins (*Silva et al., 2011*), fibronectin leucine-rich transmembrane proteins (FLRTs) (*O'Sullivan et al., 2012*), and contactins (*Zuko et al., 2016*). In the central nervous system, latrophilins have been implicated in neuronal migration, circuit assembly, and synapse formation (*Anderson et al., 2017*; *Del Toro et al., 2020*; *Donohue et al., 2021*; *Pederick et al., 2021*; *Sando et al., 2019*; *Sando and Südhof, 2021*). In humans, polymorphisms in *LPHN3* are associated with an increased risk of attention-deficit/hyperactivity disorder, and a missense variant in the *LPHN2* gene is responsible for extreme microcephaly (*Arcos-Burgos et al., 2010*; *Domené et al., 2011*; *Vezain et al., 2018*).

We recently showed in mice that one of the three latrophilins, Lphn2, displays expression patterns inverse to teneurin-3 (Ten3) in two parallel hippocampal networks (*Pederick et al., 2021*). While hippocampal *Lphn2* is preferentially expressed in the distal CA1 and the proximal subiculum, *Ten3* is enriched in the proximal CA1 and the distal subiculum. These expression patterns and reciprocal repulsions mediated by Ten3-Lphn2 interactions instruct proximal CA1 axons to target the distal subiculum, and more distal CA1 axons to target more proximal subiculum (*Figure 1A*). Specifically, Lphn2 acts as a 'receptor' in more distal CA1 axons that is repelled by Ten3 expressed from the distal subiculum (*Figure 1B*). At the same time, Lphn2 acts as a repulsive 'ligand' in the proximal subiculum to repel Ten3-expressing (Ten3+) proximal CA1 axons; this action requires Lphn2's teneurin-binding domain but not its FLRT-binding activity (*Figure 1C*; *Pederick et al., 2021*). Therefore, Lphn2 is required cell autonomously as a receptor in more distal CA1 axons for their precise target selection, and non-autonomously in target neurons as a ligand for precise target selection of proximal CA1 axons. While Ten3 additionally mediates homophilic attraction (*Berns et al., 2018*; *Pederick et al., 2021*), Lphn2 does not mediate homophilic binding in trans (*Boucard et al., 2014*; *Pederick et al., 2021*).

Structurally, the N-terminal extracellular domain of latrophilins comprises a rhamnose-binding lectin (RBL) domain, an olfactomedin-like (OLF) ligand-binding domain, a serine/threonine-rich region and

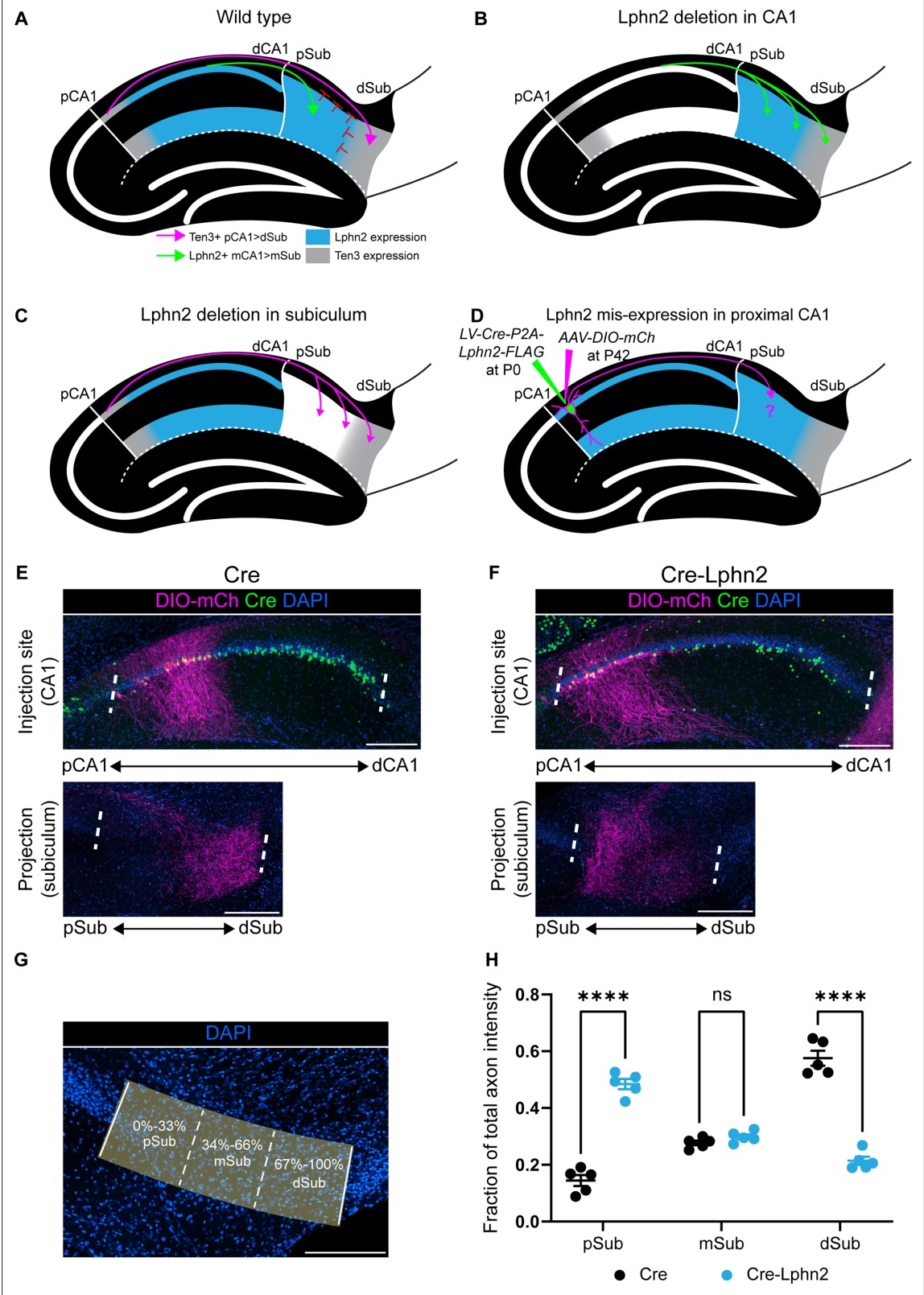

**Figure 1.** Misexpression of latrophilin-2 (Lphn2) in proximal CA1 axons causes axon mistargeting to the proximal subiculum. (**A**) Cartoon depicting the topographic connections from proximal CA1 (pCA1) to distal subiculum (dSub) and distal CA1 (dCA1) to proximal subiculum (pSub). Ten3+ proximal CA1 axons are repelled from Lphn2 expressing (Lphn2+) proximal subiculum and Lphn2+ axons are repelled from Ten3+ distal subiculum. Red symbols indicate the repulsive cues experienced by CA1 axons, previously described in *Pederick et al., 2021*. (**B**) Deletion of Lphn2 from CA1 leads to distal

*Figure 1 continued on next page*

*Figure 1 continued*

CA1 axons mistargeting to distal subiculum, suggesting that Lphn2 acts cell-autonomously as a repulsive receptor. (**C**) Deletion of Lphn2 from proximal subiculum results in proximal CA1 axon mistargeting to proximal subiculum, suggesting Lphn2 acts cell-non-autonomously as a repulsive ligand. Figures (**A–C**) are based on *Pederick et al., 2021*. (**D**) Experimental design of Lphn2 misexpression assay in proximal CA1. At postnatal day (P) 0, lentivirus expressing Cre or Cre and Lphn2 was injected into CA1. This was followed by injection at P42 of Cre-dependent membrane bound mCherry (mCh) into proximal CA1 as an axon tracer. (**E and F**) Representative images of AAV-DIO-mCh (magenta; mCh expression in a Cre-dependent manner) injections in proximal CA1 (top) and corresponding projections in the subiculum (bottom). (**G**) A representative image of the subiculum with proximal subiculum (pSub), mid subiculum (mSub), and distal subiculum (dSub) regions highlighted. (**H**) The fraction of total axon intensity within proximal, mid, and distal subiculum. *Cre*: N=5 and *Cre-Lphn2*: N=5. Means ± SEM; two-way ANOVA with Sidak's multiple comparisons test. Injection sites of all subjects are shown in *Figure 1—figure supplement 3*. Scale bars represent 200 μm.

The online version of this article includes the following source data and figure supplement(s) for figure 1:

**Source data 1.** Misexpression of latrophilin-2 (Lphn2) in proximal CA1 axons causes axon mistargeting to the proximal subiculum.

**Figure supplement 1.** In vivo expression of lentivirus used in *Figures 1, 3 and 4*.

**Figure supplement 2.** Quantification of latrophilin-2 (Lphn2), Lphn2_F831A/M835A, and Lphn2_T829G expression in CA1.

**Figure supplement 2—source data 1.** Quantification of latrophilin-2 (Lphn2), Lphn2_F831A/M835A, and Lphn2_T829G expression in CA1.

**Figure supplement 3.** Mean injection site positions for proximal CA1 axon tracing in *Figures 1, 3 and 4*.

hormone receptor motif (HRM), and a conserved GPCR autoproteolysis-inducing (GAIN) domain that encompasses the GPCR proteolysis site (GPS) (*Araç et al., 2012*; *Moreno-Salinas et al., 2019*; *Vizurraga et al., 2020*; *Figure 2A*). aGPCRs undergo autoproteolytic cleavage at the HL/T consensus site within the GPS. This self-cleavage divides the receptor into an extracellular N-terminal fragment (NTF) and a membrane-bound C-terminal fragment (CTF) that remain noncovalently associated throughout biosynthesis and membrane trafficking (*Vizurraga et al., 2020*). The seven residues immediately C-terminal to the GPS constitute the tethered agonist peptide (also known as the Stachel or stalk peptide), which upon exposure binds within the transmembrane domain to activate heterotrimeric G proteins (*Liebscher and Schöneberg, 2016*).

While our previous in vivo work established that interaction between Ten3 and Lphn2 was required for precise circuit assembly (*Pederick et al., 2021*), it did not examine how this might depend on Lphn2-mediated signaling mechanisms. Here, we modified our previous hippocampal model to develop an Lphn2 misexpression assay (*Figure 1D*). We misexpressed Lphn2 in either CA1 axons or the subiculum target and assessed the impact on normal proximal CA1→distal subiculum axon targeting. We found that ectopically expressing wild-type Lphn2 in proximal CA1 axons causes their mistargeting to the proximal subiculum. This provided us with a robust platform to interrogate whether tethered agonist activity or autoproteolytic cleavage is required for axon mistargeting in this Lphn2 ectopic expression system. When misexpressed in CA1, Lphn2 tethered-agonist activity was required for Lphn2-mediated axon mistargeting. By contrast, when we misexpressed Lphn2 in subiculum target neurons, both tethered agonist activity and autoproteolysis were dispensable for Lphn2-mediated axon repulsion. Thus, our data support that Lphn2 G-protein coupling is required in axons but not target neurons during precise circuit assembly.

## Results

### Misexpression of wild-type Lphn2 in proximal CA1 leads to axon mistargeting in the subiculum

To investigate the role of Lphn2-mediated G protein activity in hippocampal axon targeting, we first designed a gain-of-function assay in which we misexpressed Lphn2 in proximal CA1 neurons. We hypothesized that this ectopic expression would cause proximal CA1 axons to avoid the Ten3+ distal subiculum and incorrectly target the proximal subiculum. If so, this platform could provide us with an assay to test Lphn2 mutants with defects in various functions to determine whether wild-type Lphn2 mistargeting is compromised.

To test our hypothesis, we used a dual injection strategy to ectopically express Lphn2 in proximal CA1 and trace its axons into the subiculum (*Figure 1—figure supplement 1*). At postnatal day 0 (P0), lentivirus expressing *Cre* (LV-Cre) (control) or Cre-Lphn2 (LV-Cre-P2A-Lphn2) was injected into proximal CA1, followed by injection of a Cre-dependent membrane-bound mCherry (AAV-DIO-mCherry)

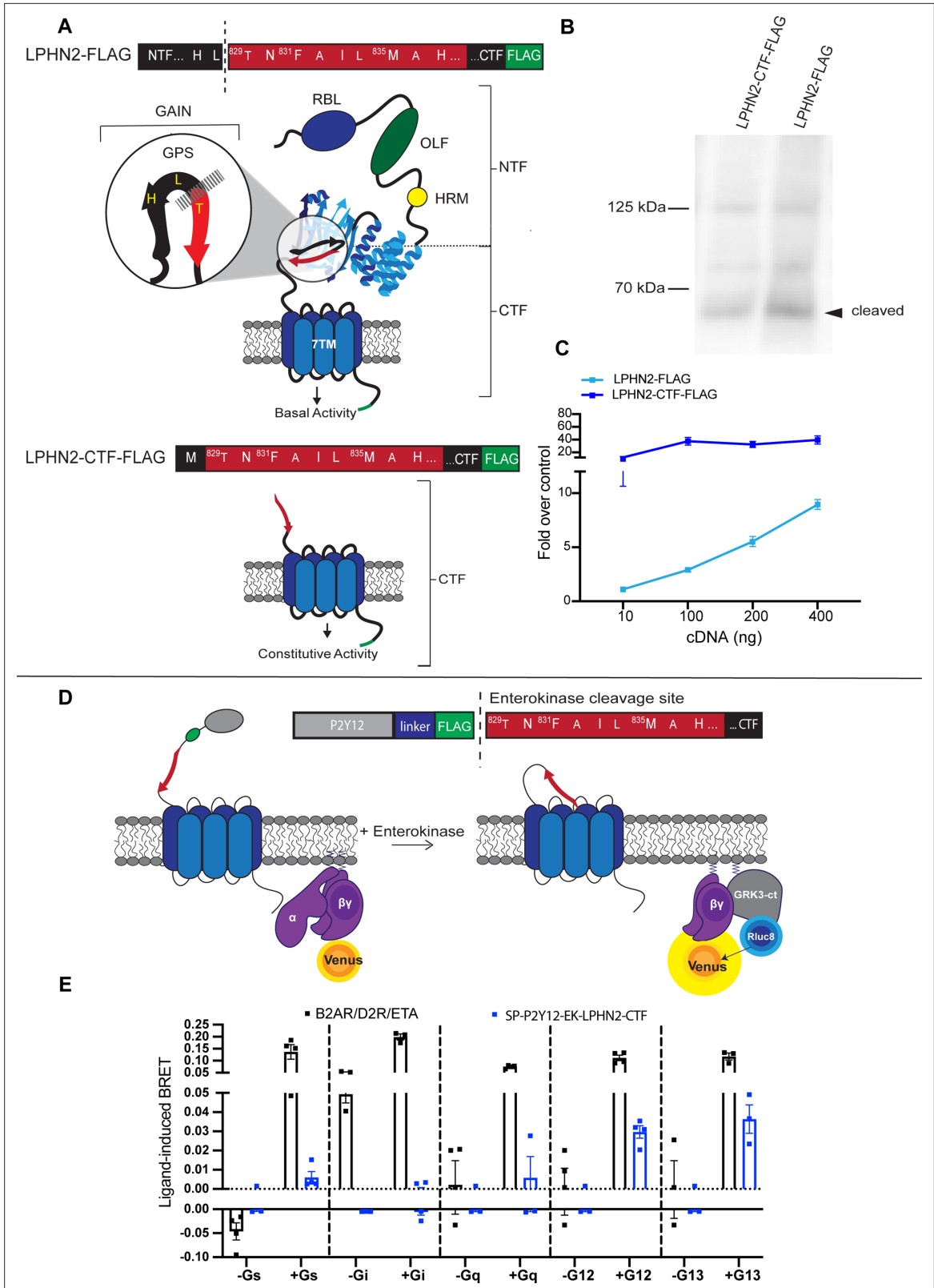

**Figure 2.** Exposure of the latrophilin-2 (Lphn2) tethered agonist (TA) promotes intracellular signaling through Gα$_{12/13}$. (**A**) Cartoon representations of full-length and tethered agonist-exposed (CTF) Lphn2 with detailed amino acid sequences for the TA. The extracellular domain of Lphn2 comprises an N-terminal rhamnose-binding lectin domain (RBL), an olfactomedin-like domain (OLF), a serine/threonine-rich region, and a HormR domain (HRM). It also contains the GPCR autoproteolysis-inducing (GAIN) domain necessary for autoproteolytic cleavage. This cleavage divides the aGPCR into two

*Figure 2 continued on next page*

*Figure 2 continued*

polypeptide chains: an N-terminal fragment (NTF) and a C-terminal fragment (CTF). The peptide stretch directly following the proteolytic cleavage site is known as the 'Stachel' or tethered agonist. Exposure of the tethered agonist results in aGPCR activation and downstream signaling. (**B**) Representative immunoblot analysis (N=3) of wild-type Lphn2 and Lphn2-CTF expression in HEK293T cells using a primary antibody against FLAG (1:500, ThermoFisher, PA1-984B). Expected bands for full-length Lphn2-FLAG and Lphn2-CTF-FLAG are 164 kDa and 72 kDa, respectively. (**C**) Serum response element (SRE) luciferase reporter assay for Lphn2 and Lphn2-CTF shows that removing the entire NTF up to the GPS cleavage site constitutively enhances SRE signaling (N=3 biological replicates, 9 technical replicates). (**D**) Schematic outlining the Gβγ-release bioluminescence resonance energy transfer (BRET) assay. The Lphn2 tethered agonist is capped with an enterokinase cleavage site (EK) preceded by a hemagglutinin signal peptide (SP), the P2Y12 N-terminal extracellular sequence, and a flexible linker (*Lizano et al., 2021*). Addition of 10 nM enterokinase generates a tethered agonist neoepitope identical to activated endogenous Lphn2. Lphn2 activation results in G protein dissociation, allowing Gβγ-Venus to associate with the C-terminus of GPCR kinase 3 (GRK3-ct) (*Hollins et al., 2009*). (**E**) Gβγ-release BRET assay testing SP-P2Y12-EK-Lphn2-CTF activation of Gα$_s$, Gα$_i$, Gα$_q$, Gα$_{12}$, and Gα$_{13}$ in HEKΔ7 cells (N=3–4 biological replicates, 9–12 technical replicates). β2-adrenergic receptor (β2AR) with 1 µM isoproteronal, dopamine receptor D2 (D2R) with 10 µM quinpirole, and endothelin receptor (ETA) with 100 nM ligand ET-1 were used as positive controls, for Gα$_s$, Gα$_i$ and Gα$_{q/12/13}$ ,respectively. Means ± SEM; Multiple unpaired t tests between no G protein and G protein conditions; **p<0.01; ****p<0.0001.

The online version of this article includes the following source data and figure supplement(s) for figure 2:

**Source data 1.** Full raw unedited blot of latrophilin-2 (Lphn2) expression in HEK293T cells.

**Source data 2.** Uncropped immunoblot analysis of latrophilin-2 (Lphn2) expression in HEK293T cells.

**Source data 3.** Replicates of the immunoblot assay.

**Figure supplement 1.** Gα$_q$-inhibitor YM-254890 does not impair serum response element (SRE) luciferase response of Lphn2-CTF.

**Figure supplement 2.** Gβγ-release bioluminescence resonance energy transfer (BRET) assay shows latrophilin-2 (Lphn2) couples to Gα$_{12}$ and Gα$_{13}$.

into proximal CA1 in the same mice at approximately P42 (*Figure 1D*). We confirmed the expression of Lphn2 in CA1 axons by the presence of FLAG immunostaining in the subiculum and that ectopic expression levels were higher than that of endogenous Lphn2 (*Figure 1—figure supplement 2A, C and D*). As expected, in control animals (Cre), Cre expressing (Cre+) proximal CA1 axons targeted the most distal parts of the subiculum (*Figure 1E*). By contrast, when Lphn2 was misexpressed in proximal CA1 (Cre-Lphn2), Cre+ proximal CA1 axons targeted the most proximal parts of the subiculum (*Figure 1F*). To analyze the location of proximal CA1 axons in the subiculum, we calculated the fraction of axon intensity within thirds of the subiculum across the proximal/distal axis (*Figure 1G*). Proximal CA1 axons misexpressing Lphn2 are located significantly more in the proximal third of the subiculum and significantly less in the distal third of the subiculum when compared to control axons (*Figure 1H*).

These data supported our hypothesis that ectopic expression of Lphn2 in proximal CA1 axons causes mistargeting to the proximal subiculum. Importantly, the phenotype observed when overexpressing Lphn2 in pCA1 axons is more severe than that observed when Ten3 is deleted (*Berns et al., 2018*), suggesting that mistargeting is not caused by disruption of Ten3 expression alone. Having established the effect of wild-type Lphn2 misexpression in proximal CA1 axons, we next sought to characterize G protein coupling of wild-type Lphn2 and generate Lphn2 mutants to test the requirement of G protein signaling in Lphn2 mediated neural circuit assembly.

## Lphn2 signals through Gα$_{12/13}$

The G protein interaction partners for Lphn2 have not been previously established. We recently showed that Lphn3, another member of the latrophilin family of aGPCRs, couples principally to Gα$_{12/13}$, and also more weakly to Gα$_q$, using a combination of gene expression assays and an activation strategy that permitted acute exposure of the tethered agonist in a live-cell system (*Mathiasen et al., 2020*). Thus, we began our signaling characterization of Lphn2 similarly using a wild-type full-length Lphn2 construct, and a constitutively active construct termed Lphn2-CTF (*Figure 2A*). The wild-type Lphn2 construct comprises all extracellular elements including the RBL, OLF, HRM, and GAIN domains, in addition to the seven transmembrane helix domain. The Lphn2-CTF lacks the entire NTF up to the GPS and instead has only a methionine residue before the tethered agonist. We tested the expression of these constructs in mammalian cells using immunoblotting and showed that both full-length Lphn2 and Lphn2-CTF ran at the expected truncated position (~72 kDa) suggesting that full-length Lphn2 undergoes normal proteolytic cleavage (*Figure 2B*). This result for full-length Lphn2 is similar to our work characterizing autoproteolysis of Lphn3 (*Perry-Hauser et al., 2022*).

To infer the activity of these constructs in G protein signaling pathways, we used a luminescence-based gene expression assay for serum response element (SRE), which produced a robust response

in our previous studies of Lphn3 (*Mathiasen et al., 2020*). In our assay design, SRE action is coupled to the transcription and translation of firefly luciferase; this readout is then normalized to the control reporter, *Renilla* luciferase, expressed from the same plasmid under a constitutive promoter. We found that Lphn2-CTF significantly enhanced signaling over wild-type Lphn2 for SRE gene expression at varying levels of cDNA transfection (*Figure 2C*). Since the SRE assay reports on signaling by $G\alpha_{12/13}$ as well as $G\alpha_q$ we tested whether $G\alpha_{12/13}$ or $G\alpha_q$ was the primary contributor to this response using a selective $G\alpha_q$ inhibitor, YM-254890 (*Figure 2—figure supplement 1*). We did not observe a significant effect upon the addition of the inhibitor, suggesting that Lphn2 signals through $G\alpha_{12/13}$.

To verify our result in the context of acute G protein activation, we next tested how tethered agonist exposure affects G protein activation in a bioluminescence resonance energy transfer (BRET) assay (*Figure 2D*). We designed a synthetically-activatable Lphn2 construct based on a recent publication that took advantage of the protease enterokinase (*Lizano et al., 2021*). Enterokinase selectively recognizes the trypsinogen substrate sequence DDDDK and cleaves after the lysine residue, thereby exposing the native tethered agonist. Thus, we cloned an Lphn2 construct that included a modified hemagglutinin signal peptide, the P2Y12 N-terminal extracellular sequence (amino acids 1–24), a flexible linker (GGSGGSGGS), the enterokinase recognition site (DYKDDDDK), and the truncated Lphn2-CTF sequence. We tested this construct in a Gβγ-release assay where energy transfer was monitored between the membrane-anchored luminescent donor, GRK3-ct-Rluc8, and the fluorescent acceptor, Gγ-Venus (*Hollins et al., 2009*). This assay was performed in a HEKΔ7 cell line with targeted deletion of $G\alpha_{12}$ and $G\alpha_{13}$, as well as $G\alpha_{s/olf}$, $G\alpha_{q/11}$, and $G\alpha_z$ (*Alvarez-Curto et al., 2016*) to enable systematic re-introduction of the Gα subunits. As expected, in the absence of Gα subunits no BRET signal was observed; however, when $G\alpha_{12}$ or $G\alpha_{13}$ was re-introduced to cells expressing the Lphn2 construct there was a significant increase in the BRET signal upon treatment with enterokinase (*Figure 2E*). This increase was not observed upon co-expression of the receptor with $G\alpha_s$, $G\alpha_{i1}$, or $G\alpha_q$ (*Figure 2E* and *Figure 2—figure supplement 2*). This suggests that the increase in cAMP reported previously for the Lphn2 CTF (*Sando and Südhof, 2021*) may not result from the direct activation of $G\alpha_s$, but rather from some other form of signaling crosstalk. Alternatively, it is possible that our *Lphn2*, which was isolated from the P8 hippocampus and lacks exons 19 and 20, may represent a different transcript variant in the brain that activates distinct signaling pathways. In fact, alternative splicing has been shown to affect G protein coupling specificity for several GPCRs, including *Lphn3* (*Markovic and Challiss, 2009*; *Röthe et al., 2019*).

Taken together, these data demonstrate that Lphn2 signals through the G proteins $G\alpha_{12}$ and $G\alpha_{13}$ in heterologous cells. Having established that these in-cell methods were sufficient to characterize G protein signaling pathways for Lphn2, we next characterized how different mutations in the tethered agonist region affect intracellular signaling.

## Mutating conserved residues F831A and M835A in the tethered agonist impairs G protein coupling activity

Previous studies suggest that the third and seventh residues of aGPCRs are required for tethered agonist-mediated G protein activation (*Stoveken et al., 2015*). We hypothesized that mutating these residues in Lphn2, phenylalanine (F831), and methionine (M835), to alanine (F831A/M835A) would impair G protein signaling mediated by the tethered agonist (*Figure 3A*). Like our work with wild-type Lphn2 and Lphn2-CTF, we mutated the tethered agonist residues in both full-length and truncated constructs (Lphn2_F831A/M835A and Lphn2-CTF_F831A/M835A, respectively). Immunoblotting against the C-terminal FLAG-tag confirmed expression in HEK293T cells but showed that Lphn2_F831A/M835A is largely uncleaved (*Figure 3B*). This is consistent with previous work with Lphn1 showing that mutating the third phenylalanine to an alanine disrupts autoproteolytic cleavage (*Araç et al., 2012*) and shows that the double mutation (F831A/M835A) in Lphn2 also inhibits cleavage. We also validated that Lphn2_F831A/M835A is expressed on the cell surface at a comparable level as Lphn2 wild-type (*Figure 3—figure supplement 1*). We then proceeded to test these constructs in our SRE gene expression system (*Figure 3A*). As hypothesized, both the full-length and truncated Lphn2 had dramatically impaired responses to SRE across varying levels of cDNA transfection.

To confirm that the reduced SRE response was due to impaired G protein coupling and not simply to impaired proteolysis, we cloned the CTF of our Lphn2_F831A/M835A mutant into our enterokinase-activatable construct. We then tested our construct in the Gβγ-release assay and compared the BRET

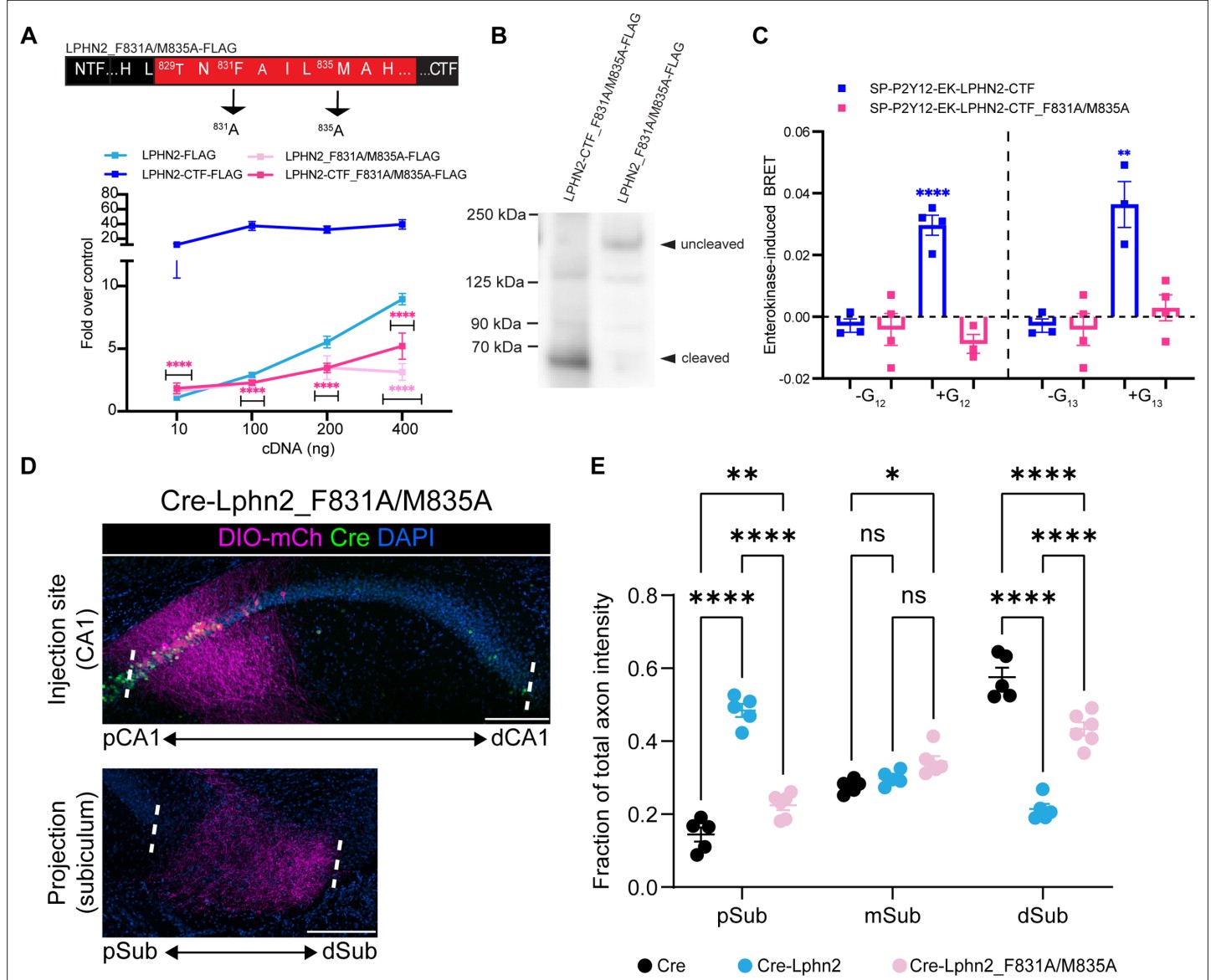

**Figure 3.** Lphn2_F831A/M835A has impaired G protein coupling activity and autoproteolytic cleavage and fails to misdirect proximal CA1 (pCA1) axons to the proximal subiculum (pSub) when misexpressed. (**A**) Schematic of the mutated tethered agonist for Lphn2_F381A/M835A. The serum response element (SRE) luciferase reporter assay shows that both the full-length Lphn2_F831A/M835A and the Lphn2-CTF_F831A/M835A have impaired signaling (N=3 biological replicates, 9 technical replicates). Means ± SEM; Multiple unpaired t-tests between full-length latrophilin-2 (Lphn2) and Lphn2_F831A/ M835A and Lphn2-CTF and Lphn2-CTF_F831A/M835A constructs; ****p<0.0001. (**B**) Representative immunoblot analysis (N=3) of TA-dead Lphn2 and TA-dead Lphn2-CTF expression in HEK293T cells using a primary antibody against FLAG (1:500, ThermoFisher, PA1-984B). Expected bands for full-length Lphn2_F831A/M835A-FLAG and Lphn2_F831A/M835A-CTF-FLAG are 164 kDa and 72 kDa, respectively. (**C**) Gβγ-release BRET assay testing SP-P2Y12-EK-Lphn2-CTF_F831A/M835A activation of Gα$_{12}$ and Gα$_{13}$ in HEKΔ7 cells (N=3–4 biological replicates, 9–12 technical replicates). SP-P2Y12-EK-Lphn2-CTF signaling is shown for comparison. Means ± SEM; Multiple unpaired t tests between no G protein and G protein conditions; *p<0.05, **p<0.01; ****p<0.0001. (**D**) Representative images of AAV-DIO-mCh (magenta; mCh expression in a Cre-dependent manner) injections in proximal CA1 (top) and corresponding projections in the subiculum (bottom). (**E**) Fraction of total axon intensity within proximal, mid, and distal subiculum. Cre: N=5, Cre-Lphn2: N=5 and Cre-Lphn2_F831A/M835A: N=6. Means ± SEM; two-way ANOVA with Sidak's multiple comparisons test. Injection sites of all subjects are shown in *Figure 1—figure supplement 3*. Scale bars represent 200 μm.

The online version of this article includes the following source data and figure supplement(s) for figure 3:

**Source data 1.** Uncropped immunoblot analysis of latrophilin-2 (Lphn2) expression in HEK293T cells.

**Source data 2.** Lphn2_F831A/M835A has impaired G protein coupling activity and autoproteolytic cleavage and fails to misdirect proximal CA1 (pCA1) axons to the proximal subiculum (pSub) when misexpressed.

*Figure 3 continued on next page*

Figure 3 continued

**Figure supplement 1.** Latrophilin-2 (Lphn2), Lphn2_F831A/M835A, and Lphn2_T829G are expressed at the cell surface at comparable levels.

**Figure supplement 2.** Comparison of fraction of total axon intensity across the subiculum within the same experimental condition.

response to wild-type Lphn2-CTF. Unlike the wild-type receptor, Lphn2-CTF_F831A/M835A did not yield a BRET signal after the re-introduction of any of the G proteins in question (G$\alpha_s$, G$\alpha_{i1}$, G$\alpha_q$, G$\alpha_{12}$, or G$\alpha_{13}$) (**Figure 3C**, **Figure 2—figure supplement 2**). Taken together, our findings demonstrate that the F831A/M835A mutations in Lphn2 impair tethered agonist-mediated G protein coupling.

## Tethered agonist activity or autoproteolysis of Lphn2 is required for its cell-autonomous effect in causing proximal CA1 axon mistargeting

We next misexpressed Lphn2-F831A/M835A in proximal CA1 to determine if Lphn2 tethered agonist activity or autoproteolysis is required in vivo to direct mistargeting of proximal CA1 axons. Lphn2_F831A/M835A was ectopically expressed at levels similar to that of wild-type Lphn2 and was detected in CA1 axons (**Figure 1—figure supplement 1A and B**). We injected LV-Cre-P2A-Lphn2_F831A/M835A-FLAG into CA1 of P0 mice, followed by AAV-DIO-mCherry into proximal CA1 of the same mice as adults. The majority of Lphn2_F831A/M835A-expressing proximal CA1 axons targeted the most distal third of the subiculum, like negative control Cre animals (**Figure 3D**). The fraction of axon intensity in Cre-Lphn2_F831A/M835A animals was significantly lower in the proximal subiculum and significantly higher in the distal subiculum when compared to Cre-P2A-Lphn2 animals (**Figure 3E**). Proximal CA1 axons in Cre-P2A-Lphn2_F831A/M835A animals showed a similar pattern of targeting to negative control Cre animals (**Figure 3—figure supplement 2**), although the total fraction of axon intensity was significantly lower in the distal subiculum (**Figure 3E**).

Collectively, these findings suggest that tethered agonist activity or autoproteolysis is required for Lphn2-mediated miswiring of proximal CA1 axons.

## Mutating residue T829G in the tethered agonist renders Lphn2 cleavage deficient but preserves the ability of the tethered agonist to activate G protein

While misexpressing Lphn2_F831A/M835A failed to cause proximal CA1 axons to mistarget to the proximal subiculum, we could not definitively link this result to impaired tethered agonist activity since the Lphn2_F831A/M835A mutant was also resistant to autoproteolytic cleavage (**Figure 3B**). Since our initial efforts to find a tethered agonist mutant with impaired G protein signaling that retained normal autoproteolytic cleavage were unsuccessful, we designed a construct that rendered Lphn2 resistant to autoproteolytic cleavage but preserved tethered agonist activity. Previous studies showed that replacing threonine-838 in the tethered agonist of Lphn1 or threonine-923 in the tethered agonist of Lphn3 to glycine inhibited autoproteolysis while maintaining proper folding of the receptor (**Araç et al., 2012**; **Kordon et al., 2023**). Thus, we mutated the analogous threonine-829 in Lphn2 (Lphn2_T829G) and confirmed that Lphn2_T829G was cleavage resistant using immunoblotting (**Figure 4A**). We also validated that Lphn2_T829G is expressed on the cell surface at a comparable level as Lphn2 wild-type (**Figure 3—figure supplement 1**).

We next assessed G protein signaling for Lphn2_T829G using our SRE gene expression system with full-length and truncated receptors (Lphn2_T829G and Lphn2-CTF_T829G, respectively) (**Figure 4B**). Full-length Lphn2_T829G had significantly impaired SRE response compared to wild-type Lphn2, consistent with diminished exposure of the tethered agonist in the absence of cleavage; however, when we tested Lphn2-CTF_T829G, which lacked the entire NTF up to the GPS cleavage site, we observed SRE levels comparable to Lphn2-CTF suggesting that the mutated tethered agonist is fully active if exposed. We, therefore, cloned the CTF of the T829G mutant into our enterokinase-activatable construct and tested BRET signaling following the re-introduction of G$\alpha$ proteins (**Figure 4C**). The T829G-CTF retained BRET signaling comparable to wild-type Lphn2-CTF for G$\alpha_{12}$ and G$\alpha_{13}$, with no discernable BRET response for G$\alpha_s$, G$\alpha_{i1}$, or G$\alpha_q$ (**Figure 2—figure supplement 2**). Taken together, these data supported that Lphn2_T829G is resistant to autoproteolytic cleavage but maintains a functional tethered agonist.

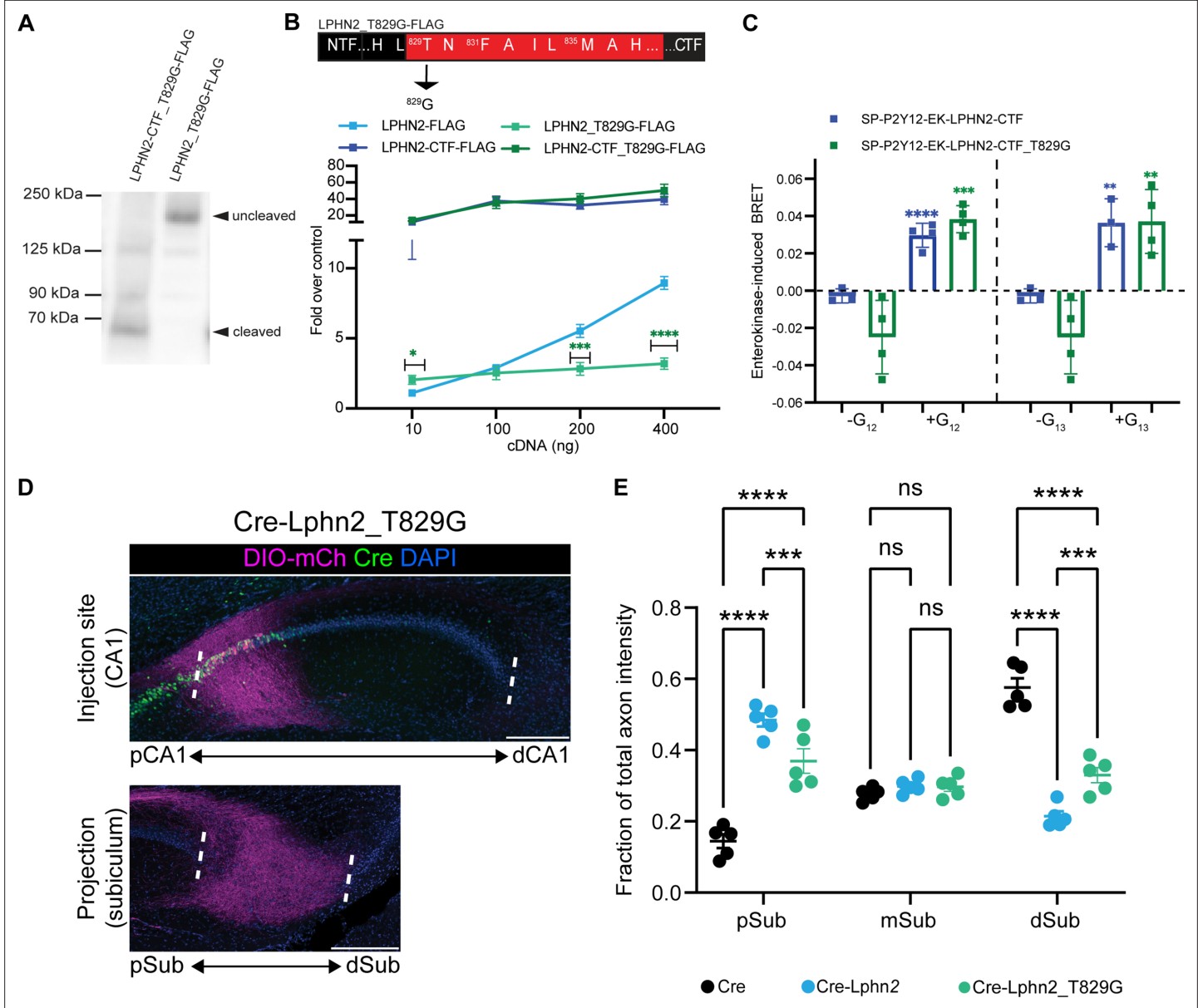

**Figure 4.** Lphn2_T829G impairs autoproteolytic cleavage, retains G protein activity in the truncated receptor, and misdirects axons to the proximal subiculum (pSub) when misexpressed. (**A**) Representative immunoblot analysis (N=3) of Lphn2_T829G and Lphn2-CTF_T829G expression in HEK293T cells using a primary antibody against FLAG (1:500, ThermoFisher, PA1-984B). Expected bands for full-length Lphn2_T829G-FLAG and Lphn2-CTF_T829G-FLAG are 164 kDa and 72 kDa, respectively. (**B**) Schematic of the mutated tethered agonist for Lphn2_T829G. The serum response element (SRE) luciferase reporter assay shows that the full-length Lphn2_T829G has impaired SRE levels while the Lphn2_T829G truncated up to the GPS cleavage site has SRE levels comparable to Lphn2_CTF (N=3 biological replicates, nine technical replicates). Means ± SEM; Multiple unpaired t tests between full-length Lphn2 and Lphn2_T829G and Lphn2-CTF and Lphn2-CTF_T829G constructs; *p<0.05; ***p<0.001; ****p<0.0001. (**C**) Gβγ-release BRET assay testing SP-P2Y12-EK-Lphn2-CTF_T829G activation of Gα$_{12}$ and Gα$_{13}$ in HEKΔ7 cells (N=3–4 biological replicates, 9–12 technical replicates). SP-P2Y12-EK-Lphn2-CTF signaling is shown for comparison. Means ± SEM; Multiple unpaired t-tests between no G protein and G protein conditions; ***p<0.001; ****p<0.0001. (**D**) Representative images of AAV-DIO-mCh (magenta; mCh expression in a Cre-dependent manner) injections in proximal CA1 (top) and corresponding projections in the subiculum (bottom). (**E**) Fraction of total axon intensity within proximal, mid, and distal subiculum. Cre: n=5, Cre-Lphn2: n=5 and Cre-Lphn2_T829G: n=5. Means ± SEM; two-way analysis of variance (ANOVA) with Sidak's multiple comparisons test. Injection sites of all subjects are shown in *Figure 1—figure supplement 3*. Scale bars represent 200 μm.

The online version of this article includes the following source data for figure 4:

**Source data 1.** Uncropped immunoblot analysis of latrophilin-2 (Lphn2) expression in HEK293T cells.

**Source data 2.** Lphn2_T829G impairs autoproteolytic cleavage, retains G protein activity in the truncated receptor, and misdirects axons to the proximal subiculum (pSub) when misexpressed.

## Autocleavage-deficient Lphn2 retains moderate activity in directing proximal CA1 axon mistargeting

To assess if autoproteolytic cleavage is required in vivo for Lphn2-mediated proximal CA1 axon mistargeting, we injected Cre-Lphn2_T829G into proximal CA1 of P0 mice, followed by AAV-DIO-mCherry into pCA1 of the same mice as adults. Lphn2_T829G was ectopically expressed at levels similar to wild-type Lphn2 and was detected in CA1 axons (*Figure 1—figure supplement 1A and B*). Overall, Lphn2_T829G-expressing proximal CA1 axons did not show highly enriched targeting to a specific region of the subiculum, as observed for control proximal CA1 axons or Lphn2 misexpressing proximal CA1 axons, which preferentially target distal and proximal subiculum, respectively (*Figure 4E*). However, compared to the control there was a significant increase in the fraction of axon intensity in the proximal subiculum in Cre-Lphn2_T829G animals, even though this mistargeting was not as pronounced as seen with wild-type Cre-Lphn2 (*Figure 3—figure supplement 2*.).

The intermediate gain-of-function phenotypes of misexpressing Lphn2_T829G compared to misexpressing wild-type Lphn2 or tethered agonist-deficient (and non-cleavable) Lphn2 suggest that autoproteolysis is not absolutely required for Lphn2 misexpression-induced miswiring of proximal CA1 axons. The weaker than wild-type overexpression phenotype is likely caused by the decreased G protein signaling of the full-length construct given the more limited exposure of the tethered agonist in the absence of cleavage. The preservation of some signaling activity of Lphn2_T829G is consistent with the ability of its tethered agonist to signal.

## Neither tethered agonist activity nor autoproteolysis is required for Lphn2's action as a repulsive ligand

We previously showed that misexpression of Lphn2 in distal subiculum target neurons causes proximal CA1 axons to avoid this area, suggesting that Lphn2 acts cell non-autonomously as a repulsive ligand in directing target selection of proximal CA1 axons (*Pederick et al., 2021*). In the context of repulsive axon guidance, proteolysis has been proposed as a mechanism to disassemble the extracellular binding complex after repulsive signaling, which is necessary for repulsion (*Hattori et al., 2000*). Are tethered agonist activity and/or autoproteolysis required for Lphn2's cell non-autonomous role in neural circuit assembly? To test this, we used a strategy to ectopically express Lphn2 in the distal subiculum and trace proximal CA1 axons into the subiculum (*Figure 5A*; *Figure 5—figure supplement 1*) identical to the one we previously reported for comparing lentiviruses expressing GFP alone (LV-GFP) or GFP and Lphn2 (LV-GFP-P2A-Lphn2-FLAG; *Pederick et al., 2021*). At postnatal day 0 (P0), LV-GFP-P2A-Lphn2_F831A/M835A-FLAG and LV-GFP-P2A-Lphn2_T829G-FLAG were injected into distal subiculum, followed by injection of membrane-bound mCherry (AAV-mCherry) into proximal CA1 in the same mice at approximately P42. The P0 lentivirus injection only covers a small fraction of the entire proximal CA1 axon projection, enabling us to assess whether proximal CA1 axons target lentivirus-expressing regions differently from adjacent regions that do not express lentivirus. To observe the relationship between proximal CA1 axon projections and lentivirus-induced regions of the subiculum, we plotted axon signal intensity (mCh) and lentivirus injection site (GFP) from the same animal as height and color, respectively.

We previously reported that GFP alone does not affect the intensity of proximal CA1 axons, whereas GFP-Lphn2 regions have significantly reduced proximal CA1 axon intensity in GFP-Lphn2 positive regions (*Pederick et al., 2021*; *Figure 5B and C*; *Figure 5—figure supplement 2A and B*). When either GFP-Lphn2_F831A/M835A or GFP-Lphn2_T829G were expressed in the distal subiculum, we also observed a significant decrease in axon intensity in GFP positive regions compared to GFP (*Figure 5D, E and F* and *Figure 5—figure supplement 2C and D*). This decrease was not significantly different from GFP-Lphn2 animals (*Figure 5F*). These findings suggest that neither tethered agonist activity nor autoproteolysis is required for Lphn2's cell-non-autonomous role as a ligand in the neural circuit assembly.

## Discussion

In this study, we utilized a combination of in vivo axon target selection and in vitro cell signaling assays to determine if Lphn2 G protein signaling is required for its role as a neural wiring molecule. First, we showed that Lphn2 misexpression can cell-autonomously misdirect proximal CA1 axons to the

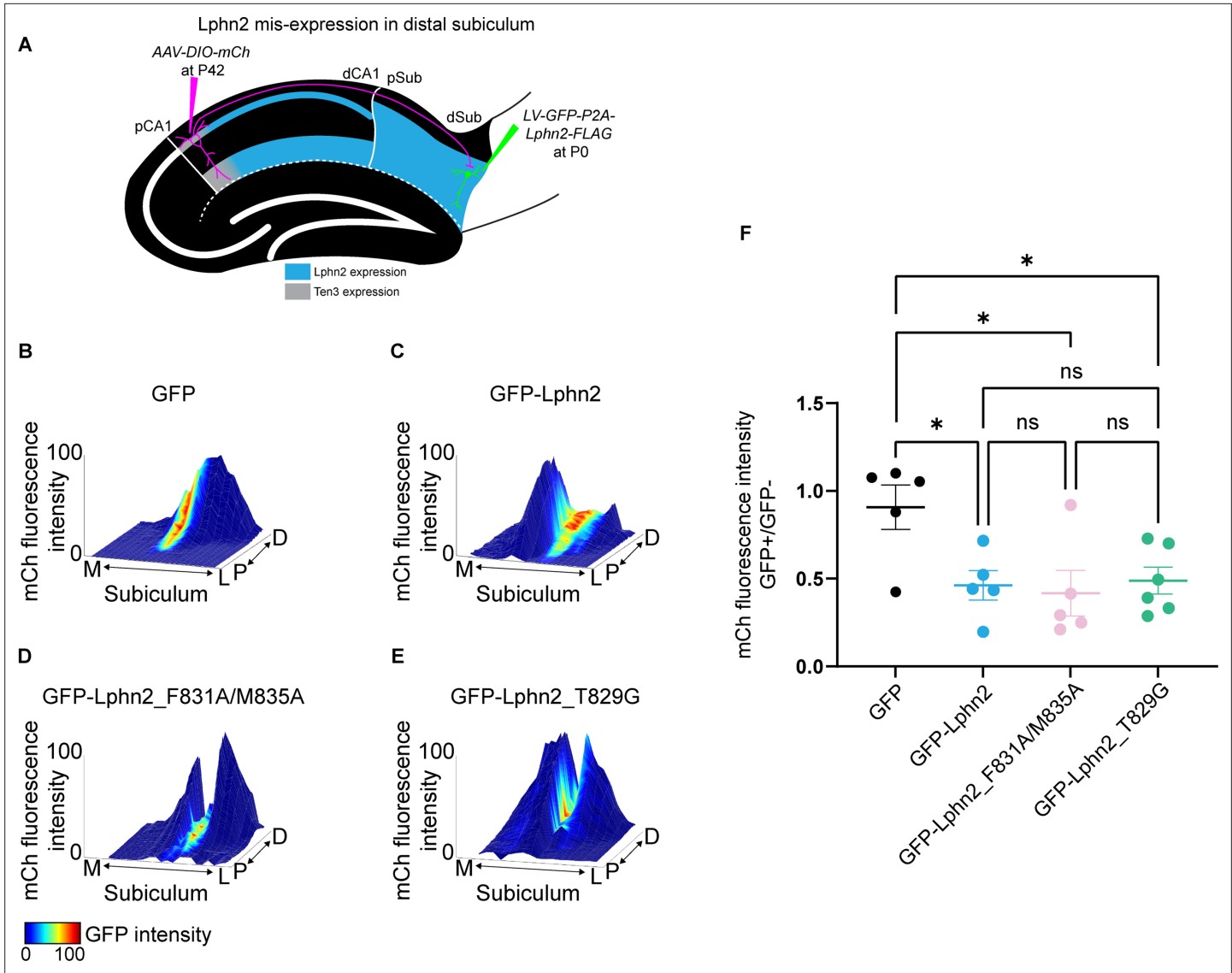

**Figure 5.** Misexpression of latrophilin-2 (Lphn2) mutants in target distal subiculum neurons does not cause mistargeting of proximal CA1 axons. (**A**) Experimental design of Lphn2 misexpression assay in distal subiculum. (**B to E**). Representative mountain plots showing normalized mCh fluorescence as height (proximal CA1 axon projections in subiculum) and normalized GFP fluorescence as color (lentivirus expression). P, proximal; D, distal; M, medial; L, lateral. (**F**) Ratio of mCh fluorescence intensity (from proximal CA1 axons) in GFP+ versus GFP– regions of the subiculum. GFP: N=5, GFP-Lphn2: N=5, GFP-Lphn2_F831A/M835A: N=5 and GFP-Lphn2_T829G: N=6. Means ± SEM. One-way analysis of variance (ANOVA) with Tukey's multiple comparisons test.

The online version of this article includes the following source data and figure supplement(s) for figure 5:

**Source data 1.** Misexpression of Lphn2 mutants in target distal subiculum neurons does not cause mistargeting of proximal CA1 axons.

**Figure supplement 1.** In vivo expression of lentivirus used in *Figure 5*.

**Figure supplement 2.** Representative images corresponding to *Figure 5B–E*.

proximal subiculum, establishing an assay to test the requirements of Lphn2 G protein signaling when it acts as a receptor (*Figure 1*). Second, we identified the G protein interaction partners of Lphn2 (*Figure 2*) and validated point mutations that disrupt tethered agonist activity and/or autoproteolysis of the GPS region (*Figures 3 and 4*). Third, we showed that when Lphn2 is misexpressed in CA1 axons, tethered agonist activity is required for Lphn2's ability to misdirect axon targeting (*Figures 3*

*and 4*). Finally, when Lphn2 acts as a repulsive ligand in subiculum target neurons, we demonstrated that neither tethered agonist activity nor autoproteolytic cleavage is required for the receptor's ability to repel Ten3+ proximal CA1 axons (*Figure 5*). Taken together, these findings highlight the importance of Lphn2 G protein signaling during precise circuit assembly in a context-specific manner. Our results also support that while aGPCR GPS cleavage is dispensable for Lphn2's role as a receptor to direct axon targeting, an intact tethered agonist is essential.

## The role of autoproteolytic cleavage and tethered agonism in aGPCR activation

Upon aGPCR biosynthesis, the conserved GAIN domain undergoes autoproteolytic cleavage at the GPS to generate N- and C-terminal fragments that remain non-covalently bound during trafficking to the cell surface (*Araç et al., 2012*). Crystal structures of GAIN domains from Lphn1 and ADGRB3 (BAI3) (*Araç et al., 2012*), ADGRG1 (GPR56) (*Salzman et al., 2016*), and ADGRG6 (GPR126) (*Leon et al., 2020*), revealed that in an intact aGPCR, the tethered agonist is buried as a β-strand in the GAIN domain, forming an extensive network of conserved hydrogen bonds and hydrophobic side chains. This suggests that when the complex between aGPCR's NTF and CTF remains intact the tethered agonist is inaccessible for engagement with its binding site in the 7TM domain. However, several studies have reported that naturally cleavage-resistant aGPCRs can still function (*Liebscher et al., 2014*; *Wilde et al., 2016*). This suggests that the tethered agonist can in fact interact with the 7TM independent of cleavage, although the efficacy of this interaction is likely to be diminished, as we see in the present study. Thus, the relative contributions and/or necessity of autoproteolytic cleavage and the tethered agonist to aGPCR activity remain an area of active study.

Recent efforts reported the structures of eight aGPCRs, seven of which were truncated up to the GPS (*Barros-Álvarez et al., 2022*; *Ping et al., 2022*; *Qu et al., 2022*; *Xiao et al., 2022*). While most discussions from the structural aGPCR studies argued that NTF dissociation is required for tethered agonist interaction with the receptor, the structure of autoproteolysis-deficient ADGRF1 supported the possibility of the cleavage-independent manner of receptor activation (*Qu et al., 2022*). The density for the tethered agonist of ADGRF1 was well-resolved and bound in an α-helical structure within the orthosteric site of the 7TM bundle. This interaction was like that of the cleaved structures, and ADGRF1 was also observed to be bound to a miniG$_{i1}$, supporting that receptor cleavage and tethered agonist exposure are not absolutely required for G protein coupling. One caveat, however, is that there is little density for the NTF in this structure, suggesting that the structure obtained may result from a fraction of receptor where cleavage has still occurred and the NTF has dissociated.

As mentioned above, not all aGPCRs are auto-proteolytically cleaved; therefore, activation cannot be fully dependent on tethered agonist exposure through the removal of the NTF (*Kishore et al., 2016*; *Liebscher et al., 2022*). In this regard, it is possible that full-length aGPCRs exist in multiple conformational states that include receptor molecules in which the tethered agonist is unmasked from the GAIN domain. In fact, molecular dynamics (MD) simulations of spontaneous tethered agonist exposure were recently reported for five intact aGPCR homologs (ADGRB3, ADGRE2, ADGRE5, ADGRG1, and Lphn1) (*Beliu et al., 2021*). Here, the authors show that tethered agonist exposure occurs due to the high intrinsic flexibility of the GAIN domain. They also used biorthogonal labeling of conserved positions within the tethered agonist to show that large portions (+6 residues) of the tethered agonist can become solvent accessible in the context of the GAIN domain. They argue that tethered agonist exposure likely occurs in a stepwise mechanism where the tethered agonist is uncovered along its N→C axis. Thus, it is possible that an intact complex of aGPCR's NTF and CTF could unmask the tethered agonist sufficiently for interaction with the 7TM, resulting in receptor activation.

The ability of tethered agonist exposure to occur in intact aGPCRs could provide an explanation for why our Lphn2_T829G mutant displays a partial axon mistargeting phenotype (*Figure 4*). Even though the Lphn2_T829G mutant cannot undergo autoproteolytic cleavage, it still retains a functional tethered agonist that is able to initiate G protein signaling if transiently unmasked. This is likely why activation of full-length Lphn2_T829G is less robust than the wild-type Lphn2, which can more readily unmask the TA. We also cannot rule out the possibility that a small amount of cleavage, although undetectable in the HEK cells immunoblotting, nonetheless contributes to the partial mistargeting phenotype in vivo.

Like the Lphn2_T829G mutant, Lphn2_F831A/M835A cannot undergo autoproteolytic cleavage. However, this mutant also has impaired G protein coupling activity even with full exposure of the TA. This explains why we observed close-to-normal axon targeting when we overexpressed Lphn2_F831A/M835A in proximal CA1 (*Figure 3*). Even if the tethered agonist of Lphn2_F831A/M835A becomes unmasked it still cannot initiate tethered agonist-mediated receptor. However, given the statistically significant difference in total axon intensity in dSub between control and Lphn2_F831A/M835A (*Figure 3E*), we cannot rule out some residual tethered agonist-independent, G protein-mediated signaling or an Lphn2 function independent of G protein signaling, at least in the context of overexpression.

### Implication of Lphn2 signaling in neural circuit assembly

How could Lphn2-mediated G protein signaling in the CA1 axons lead to axon repulsion? We show here that Lphn2 primarily signals through $G\alpha_{12}$ and $G\alpha_{13}$ in heterologous cells (*Figure 2E*). $G\alpha_{12}/G\alpha_{13}$ are known to regulate Rho GTPase; for example, $G\alpha_{13}$ binds to and activates p115RhoGEF, an exchange factor for and activator of the small GTPase RhoA (*Kozasa et al., 2011*). RhoA activation is known to cause growth cone collapse and neuronal process retraction via its regulation of the actin-myosin contractility (*Luo, 2002*; *Spillane and Gallo, 2014*). Thus, if Lphn2 coupling to $G\alpha12/G\alpha13$ also applies to CA1 neurons, as suggested by our results in heterologous cells, $G\alpha12/G\alpha13 \rightarrow RhoGEF \rightarrow RhoA$ may be a plausible pathway for Lphn2 to mediate its function as a receptor for axon repulsion.

Interestingly, neither autoproteolytic cleavage nor tethered agonist activity is required for Lphn2 to act cell non-autonomously as a repulsive ligand in subiculum target neurons (*Figure 5*). This suggests that cleavage of Lphn2 is not required for repulsion in this context and implies that another mechanism mediates the disassembly of the extracellular binding complex, which is required for retracting axons to pull away from the targets. Other potential mechanisms to disassemble the extracellular binding complex include Ten3 cleavage (teneurins are known to also undergo proteolytic cleavage at its extracellular domain; *Sita et al., 2019*), endocytosis of the adhesion complex as in the case of ephrin/Eph receptor (*Egea and Klein, 2007*), or forces produced by actin-myosin contractility in axon terminals induced by repulsive signaling. Indeed, since Lphn2 also acts as a repulsive ligand and Ten3 as a repulsive receptor, extracellular binding of Lphn2 to Ten3 should also trigger a repulsive response in Ten3+ axon terminals, but the signaling mechanism is completely unknown. Future studies on the mechanisms that disassemble the extracellular complex and intracellular signaling in the axon downstream of Ten3 will increase our understanding of how the interaction of these two molecules can lead to repulsive outcomes.

## Materials and methods

### Key resources table

| Reagent type (species) or resource | Designation | Source or reference | Identifiers | Additional information |
|---|---|---|---|---|
| Cell line (*Homo sapiens*) | HEK293T (epithelial, kidney) | ATCC | RRID:CVCL_0063 | |
| Cell line (*Homo sapiens*) | HEKΔ7 | *Alvarez-Curto et al., 2016*, PMCID:PMC5207144 | *JBC* | HEK293 cells with targeted deletion via CRISPR-Cas9 of *GNAS, GNAL, GNAQ, GNA11, GNA12, GNA13, and GNAZ* |
| Antibody | anti-FLAG (rabbit polyclonal) | ThermoFisher, PA1-984B | RRID:AB_347227 | IB: 1:500 |
| Antibody | anti-rabbit HRP (donkey polyclonal) | ThermoFisher, Cat #31458 | RRID:AB_228213 | IB: 1:10,000 |
| Recombinant DNA reagent | SRE-luciferase/glo (plasmid) | *Nazarko et al., 2018*, PMCID:PMC6137404 | *iScience* | |
| Recombinant DNA reagent | Lphn2-Flag (plasmid) | This paper | | pCDNA3.1 with Kozak (GCC) and C-terminal Flag tag |
| Recombinant DNA reagent | Lphn2-CTF-Flag (plasmid) | This paper | | pCDNA3.1 with Kozak (GCC), Lphn2 C-terminal fragment and C-terminal Flag |
| Recombinant DNA reagent | SP-P2Y12-EK-Lphn2-CTF (plasmid) | This paper | | Enterokinase cleavage site based on *Lizano et al., 2021*, Lphn2-CTF |
| Recombinant DNA reagent | Lphn2-F831A/M835A-Flag (plasmid) | This paper | | pCDNA3.1 with kozak (GCC), Lphn2, F831A and M835A, and C-terminal Flag |

*Continued on next page*

*Continued*

| Reagent type (species) or resource | Designation | Source or reference | Identifiers | Additional information |
|---|---|---|---|---|
| Recombinant DNA reagent | Lphn2-F831A/M835A-CTF-Flag (plasmid) | This paper | | pCDNA3.1 with kozak (GCC), Lphn2, F831A and M835A C-terminal fragment, and C-terminal Flag |
| Recombinant DNA reagent | SP-P2Y12-EK-Lphn2-F831A/M835A-CTF (plasmid) | This paper | | Enterokinase cleavage site based on *Lizano et al., 2021*, Lphn2-F831A/M835A-CTF |
| Recombinant DNA reagent | Lphn2-T829G-Flag (plasmid) | This paper | | pCDNA3.1 with kozak (GCC), Lphn2, T829G, and C-terminal Flag |
| Recombinant DNA reagent | Lphn2-T829G-CTF-Flag (plasmid) | This paper | | pCDNA3.1 with kozak (GCC), Lphn2, T829G C-terminal fragment, and C-terminal Flag |
| Recombinant DNA reagent | SP-P2Y12-EK-Lphn2-T829G-CTF (plasmid) | This paper | | Enterokinase cleavage site based on *Lizano et al., 2021*, Lphn2-T829G-CTF |
| Recombinant DNA reagent | Gαs/Gαi/ Gαq/ Gα12/Gα13 (plasmid) | *Hollins et al., 2009*, PMCID:PMC2668204 | *Cell Signal* | |
| Recombinant DNA reagent | Gβ1 | *Hollins et al., 2009*, PMCID:PMC2668204 | *Cell Signal* | |
| Recombinant DNA reagent | Gγ2-Venus | *Hollins et al., 2009*, PMCID:PMC2668204 | *Cell Signal* | |
| Recombinant DNA reagent | GRK3ct-Rluc8 | *Hollins et al., 2009*, PMCID:PMC2668204 | *Cell Signal* | |
| Chemical compound, drug | Firefly D-luciferin | NanoLight Technology | Cat #306 | |
| Chemical compound, drug | Coelenterazine-h | NanoLight Technology | Cat #301 | |
| Chemical compound, drug | YM-254890 | AdipoGene Life Sciences | CAS 568580-02-9 | |
| Chemical compound, drug | Janelia Fluor 646 | Lavis Lab, Howard Hughes Medical Institute Janelia Research Campus | | |
| Genetic reagent (*Mus musculus*) | CD-1 | Charles River Laboratory | | CD-1 mice were used for all animal experiments |
| Recombinant DNA reagent | Lentiviral UbC-Cre | This paper | | Lentivirus co-expressing Cre for in vivo experiments |
| Recombinant DNA reagent | Lentiviral UbC-Cre-P2A-Lphn2-FLAG | This paper | | Lentivirus co-expressing Cre-P2A-Lphn2-FLAG for in vivo experiments |
| Recombinant DNA reagent | Lentiviral UbC-Cre-P2A Lphn2_F831A/M835A-FLAG | This paper | | Lentivirus co-expressing Cre-P2A-Lphn2_F831A/M835A-FLAG for in vivo experiments |
| Recombinant DNA reagent | Lentiviral UbC-Cre-P2A-Lphn2_T828G-FLAG | This paper | | Lentivirus co-expressing Cre-P2A-Lphn2_T829G-FLAG for in vivo experiments |
| Recombinant DNA reagent | Lentiviral UbC-GFP | *Pederick et al., 2021*, PMCID:PMC8830376 | *Science* | Lentivirus co-expressing GFP for in vivo experiments |
| Recombinant DNA reagent | Lentiviral UbC-GFP-P2A-Lphn2-FLAG | *Pederick et al., 2021*, PMCID:PMC8830376 | *Science* | Lentivirus co-expressing GFP-P2A-Lphn2-FLAG for in vivo experiments |
| Recombinant DNA reagent | Lentiviral UbC-GFP-P2A-Lphn2_F831A/M835A-FLAG | This paper | | Lentivirus co-expressing GFP-P2A-Lphn2_F831A/M835A-FLAG for in vivo experiments |
| Recombinant DNA reagent | Lentiviral UbC-GFP-P2A-Lphn2_T828G-FLAG | This paper | | Lentivirus co-expressing GFP-P2A-Lphn2_F831A/M835A-FLAG for in vivo experiments |
| Recombinant DNA reagent | AAV8-EF1a-DIO-ChR2-mCh | Addgene plasmid 20297 | | AAV used to label axons of Cre expressing neurons with mCherry |
| Recombinant DNA reagent | AAV8-CaMKIIa-ChR2-mCh | Addgene plasmid26975 | | AAV used to label axons of neurons with mCherry |
| Chemical compound, drug | DAPI | ThermoFisher | D1306 | 1:10,000 |
| Antibody | anti-mCherry (rat monoclonal) | ThermoFisher, M11217 | RRID:AB_2536611 | Immunohistochemistry 1:1,000 |
| Antibody | anti-Cre (Rabbit polyclonal) | Synaptic Systems, 257 003, | RRID:AB_2619968 | Immunohistochemistry 1:500 |
| Antibody | anti-GFP (chicken polyclonal) | Aves Labs, GFP-1020 | RRID:AB_10000240 | Immunohistochemistry 1:2,500 |
| Antibody | anti-FLAG (goat polyclonal) | Abcam, ab95045 | RRID:AB_10676074 | Immunohistochemistry 1:3000 |
| Antibody | anti-Lphn2 (Rabbit polyclonal) | Novus Biologicals, nbp2-58704 | | Immunohistochemistry 1:500 |

*Continued on next page*

*Continued*

| Reagent type (species) or resource | Designation | Source or reference | Identifiers | Additional information |
|---|---|---|---|---|
| Software, algorithm | Zen | Zeiss | | Previously existing |
| Software, algorithm | ImageJ | National Institutes of Health | | Previously existing |
| Software, algorithm | Adobe Photoshop | Adobe | | Previously existing |
| Software, algorithm | Adobe Illustrator | Adobe | | Previously existing |
| Software, algorithm | Matlab | MathWorks | | Previously existing |
| Software, algorithm | GraphPad Prism 9 | GraphPad Software | | Previously existing |

## Materials for cell culture experiments

Dulbecco's Modified Eagle Medium (DMEM), high glucose, and penicillin-streptomycin (P/S) (10,000 U/mL) were purchased from Gibco (ThermoFisher Scientific, Waltham, MA). Fetal bovine serum (FBS), 0.5% trypsin, and Dulbecco's phosphate-buffered saline (DPBS) were purchased from Corning (Fisher Scientific, Waltham, MA). Opti-MEM reduced serum medium, no phenol red, and Lipofectamine 2000 transfection reagent was purchased from Invitrogen (ThermoFisher Scientific). FuGENE transfection reagent was purchased from Promega (Madison, WI). RIPA buffer was purchased from Sigma-Aldrich (St. Louis, MO). Triton lysis buffer consisted of 0.11 M Tris-HCl powder, 0.04 M Tris-base powder, 75 mM NaCl, 3 mM $MgCl_2$, and 0.25% Triton X-100 pure liquid. The 3 X Firefly Assay Buffer was freshly prepared in Triton lysis buffer and contained 15 mM DTT, 0.6 mM coenzyme A (MedChemExpress, Monmouth Junction, NJ), 0.45 mM ATP (MedChemExpress, Monmouth Junction, NJ), and 0.42 mg/mL firefly D-luciferin (NanoLight Technology). *Renilla* Salts buffer consisted of 45 mM $Na_2EDTA$, 30 mM Na Pyrophosphate, and 1.425 M NaCl. The 3 X *Renilla* Assay Buffer was freshly prepared in *Renilla* Salts and contained 0.06 mM PTC124 in DMSO (MedChemExpress) and 0.01 mM coelenterazine-*h* (NanoLight Technologies, Pinetop, AZ). For the BRET assays, enterokinase, light chain, was obtained from New England Biolabs (Ipswich, MA), isoproterenol and quinpirole from Sigma Aldrich, and endothelin 1 (ET-1) from Tocris Bioscience (Bristol, United Kingdom). YM-254890 was purchased from AdipoGen Life Sciences (San Diego, CA). Impermeant Janelia Fluor 646 conjugated to benzyl guanine was a kind gift from Dr. Luke Lavis (Howard Hughes Medical Institute Janelia Research Campus).

## Plasmid DNA constructs

*Lphn2* was amplified from cDNA isolated from the P8 mouse hippocampus. Sanger sequencing confirmed that exons 19 and 20 were excluded from the amplified *Lphn2* (Refer to NCBI Reference Sequence: NM_001081298.2 for exon annotation). This cDNA was used as a polymerase chain reaction (PCR) template to make the *Lphn2* constructs used in this study. All cDNA constructs were assembled in a pCDNA3.1+ vector by Gibson assembly using NEBuilder HiFi DNA Assembly Master Mix (New England Biolabs). Sequences were confirmed with the Genewiz sequencing service (South Plainfield, NJ). Plasmid DNA constructs are available upon request.

## Cell culture

HEK293T cells (American Type Culture Collection, Manassas, VA; RRID:CVCL_0063) and HEK293 cells with targeted deletion via CRISPR-Cas9 of *GNAS, GNAL, GNAQ, GNA11, GNA12, GNA13, and GNAZ* (HEKΔ7) (*Alvarez-Curto et al., 2016*) were maintained in high-glucose DMEM supplemented with 10% FBS and 1% P/S at 37 °C in a 5% $CO_2$ humidified incubator. Cell authentication was not performed as the cells were obtained directly from the supplier. PCR-based mycoplasma testing was performed routinely using ATCC mycoplasma testing services. Cell viability was assessed for each passage using the Countess II automated cell counter (ThermoFisher Scientific).

## Immunoblot analysis

HEK293T cells were detached for 2–3 min using 0.5% trypsin and then plated at a density of 350,000 cells/mL in a six-well culture plate. After 24 hr, the cells were transfected using FuGENE transfection reagent (8 µL/2 µg cDNA) and Opti-Mem with receptor cDNA (2 µg). After 24 hr, cells were placed on ice and incubated in 500 µL RIPA buffer for 30 min. Following this incubation, cells were

scraped from the culture plate and moved into 1.5 mL microcentrifuge tubes. Cells were then spun at 15,000 × g in a 4 °C benchtop centrifuge to pellet debris. After centrifugation, 50 µL of the supernatant was transferred into a fresh microcentrifuge tube and combined with 50 µL 2 X SDS Laemmli sample buffer (Sigma-Aldrich). In preparation for immunoblot analysis, a 20 µL sample was run on an SDS-PAGE gel (Mini-PROTEAN TGX, 4–15%, Bio-Rad Laboratories, Inc, Hercules, CA) prior to transfer to a PDVF membrane (Immobilon-P Membrane, Merck Millipore Ltd., Burlington, MA). The membrane was then incubated in a 5% milk tris-buffered saline with 0.1% tween-20 (TBS-T) solution for 1 hr at RT with gentle rotation. The membrane was washed five times 5 min in TBS-T prior to overnight incubation at 4 °C with 1° anti-FLAG antibody (1:500, ThermoFisher, PA1-984B; RRID:AB_347227). The next morning, the membrane was washed five times 5 min in TBS-T. The membrane was then incubated for 1 hr at RT with 2° anti-rabbit HRP antibody (1:10,000, ThermoFisher, Cat #31458;RRID:AB_228213). The membrane was washed five times 5 min in TBS-T prior to visualization with SuperSignal West Pico Chemiluminescent Substrate (Fisher Scientific) using the Azure c600 Gel Imaging System (Azure Biosystems, Dublin, CA).

## Gene expression assays

In preparation for transfection, HEK293T cells were detached for 2–3 min using 0.5% trypsin and then seeded at a density of 400,000 cells/mL in a 12-well culture plate. After 24 hr, the cells were co-transfected using Lipofectamine 2000 (2.5 µL/1 µg cDNA) and Opti-Mem with receptor cDNA (10–600 ng), gene reporter cDNA (600 ng), and empty vector pCDNA5/FRT to balance the total amount of cDNA up to 1200 ng. After 6 hr with the transfection reagent, the media was volume exchanged to serum-free DMEM supplemented with 1% P/S (~18 hr serum starvation).

After 24 hr, the media was aspirated from the cells and each well was gently rinsed with DPBS. Cells were then mechanically detached using 275 µL DPBS and 80 µL of the resuspension was distributed in triplicate to a 96-well black/white isoplate (Perkin Elmer Life Sciences). Next, 40 µL of 3 X Firefly Assay Buffer was added to each well. The emission was then read at 535 nm after 10 min incubation using a PHERAstar FS microplate reader (BMG LABTECH, Ortenberg, Germany). Next, 60 µL 3 X *Renilla* Assay Buffer was added to each well. The emission was then read at 475 nm after 10 min incubation using a PHERAstar FS microplate reader. For assays using the Gαq-inhibitor YM-254890, the cell media was exchanged to DMEM containing 1 µM YM-254890 approximately 6 hr after transfection.

## Bioluminescence resonance energy transfer assays

In preparation for transfection, HEKΔ7 cells were detached for 2–3 min using 0.5% trypsin and then seeded at a density of 400,000 cells/mL in a 12-well culture plate. After 24 hr, the cells were co-transfected using Lipofectamine 2000 (2.5 µL/1 µg cDNA) and Opti-Mem with receptor cDNA (200 ng), Gα (720 ng), Gβ1 (250 ng), Gγ2-Venus (250 ng), membrane-anchored GRK3ct-Rluc8 (50 ng), and empty vector pCDNA5/FRT to balance the total amount of cDNA up to 1470 ng. After 24 hr transfection, cells were washed with DPBS before being re-suspended in 400 µL BRET buffer (DPBS containing 5 mM glucose). Next, 45 µL of the resuspension was distributed to six wells of a 96-well OptiPlate black-white plate (Perkin Elmer Life Sciences, Waltham, MA). Cells were then incubated for 10 mins with 10 µL coelenterazine-*h* (final concentration 5 µM) before ligand addition to reach a final well volume of 100 µL. Donor (Rluc8) and acceptor (mVenus) emission was read using a PHERAstar FS microplate reader at 485 nm and 525 nm, respectively. The BRET ratio was then measured by dividing the 525 emissions by the 485 emissions. The drug-induced BRET ratio was then calculated by subtracting the buffer BRET for each condition.

## Surface expression measurements using SNAPfast-tag

In preparation for transfection, HEK293T cells were detached for 2–3 min using 0.5% trypsin and then seeded at a density of 900,000 cells/well in a six-well culture plate. After 24 hr, the cells were transfected using FuGENE transfection reagent (8 µL/2 µg cDNA) and Opti-Mem with SNAPfast-tagged receptor cDNA (2 µg). After 24 hr, cells were incubated for 30 min with 500 µL 1 µM impermeant Janelia Fluor 646 conjugated to benzyl guanine was dissolved in DMEM containing 10% FBS and 1% P/S. Cells were then washed three times with complete DMEM and once with DPBS prior to resuspension in 500 µL DPBS. Next, 100 µL of resuspension was added to three wells of a 96-well OptiPlate black plate (Perkin Elmer Life Sciences, Waltham, MA). The emission was then read using

the filter 640/680 at a gain of 1000 using a PHERAstar FS microplate reader (BMG LABTECH, Orten-berg, Germany).

## Mice

All procedures followed animal care and biosafety guidelines approved by Stanford University's Administrative Panel on Laboratory Animal Care (APLAC 14007) and Administrative Panel on Biosafety (APB-3669-LL120) in accordance with NIH guidelines. Both male and female mice were used, and mice were group housed on a 12 hr light/dark cycle with access to food and water ad libitum. CD-1 mice from Charles River Laboratories were used for all experiments. The total number of mice injected and screened for each experiment is as follows: *Figure 1*: LV-Cre, 101, LV-Cre-P2A-Lphn2-FLAG, 60; *Figure 3*: LV-Cre-P2A-Lphn2_F831A/M835A-FLAG, 86; *Figure 4*: LV-Cre-P2A-Lphn2_T829G-FLAG, 102; and *Figure 5*: LV-GFP-P2A-Lphn2_F831A/M835A-FLAG, 79 and LV-GFP-P2A-Lphn2_T829G-FLAG, 56.

## Lentivirus generation

All lentivirus constructs expressing Cre, GFP, Lphn2, Lphn2_F831A/M835A, or Lphn2_T829G were made by inserting corresponding cDNA into the LV-UbC plasmid (*Pederick et al., 2021*) with a P2A sequence between the two ORFs. Cre and GFP were amplified from LV-UbC-GFP-Cre and full-length *Lphn2* cDNA was isolated from a cDNA library made from mRNAs from the P8 mouse hippocampus. GFP and Lphn2 were inserted into LV-UbC with a Gibson assembly cloning kit (NEB E5510S). The Lphn2_F831A/M835A and Lphn2_T829G mutations were made using Q5 mutagenesis (NEB, E0552S). All plasmids were sequenced and verified before the virus was produced. All custom lentiviruses were generated by transfecting 36 10 cm plates (HEK293T) with four plasmids (4.1 R, RTR2, VSVg, and transfer vector containing gene of interest). Medium was collected 48 hr later and centrifuged at 8400 relative centrifugal force (rcf) for 18 hr at 4 °C. Viral pellets were dissolved with PBS and further purified with a 20% sucrose gradient centrifugation at 80,000 rcf for 2 hr.

## Stereotactic injections in neonatal mice

P0 mice were anesthetized using hypothermia. CA1 injections were 1.0 mm lateral, 0.85 mm anterior, and 0.8 mm ventral from lambda and subiculum injections were 1.3 mm lateral, 0.45 mm anterior, and 0.8 mm ventral from lambda. 100 nl of lentivirus was injected at 100 nl/min at the following titers: LV-Cre ($7 \times 10^{12}$ copies per ml), LV-Cre-P2A-Lphn2-FLAG ($2.4 \times 10^{12}$ copies per ml), LV-Cre-P2A-Lphn2_F831A/M831A-FLAG ($2.24 \times 10^{12}$ copies per ml) and LV-Cre-P2A-Lphn2_T829G-FLAG ($1.2 \times 10^{13}$ copies per ml), LV-GFP ($6 \times 10^{12}$ copies per ml), LV-GFP-P2A-Lphn2-FLAG ($5 \times 10^{12}$ copies per ml), LV-GFP-P2A-Lphn2_F831A/M831A-FLAG ($3.6 \times 10^{12}$ copies per ml), LV-GFP-P2A-Lphn2_T829G-FLAG ($9 \times 10^{12}$ copies per ml).

## Stereotactic injection in adult mice

Injections of AAV8-EF1a-DIO-ChR2-mCh ($2 \times 10^{12}$ copies per ml, Neuroscience Gene Vector and Virus core, Stanford University) and AAV8-CaMKIIa-ChR2-mCh ($2 \times 10^{12}$ copies per ml, Neuroscience Gene Vector and Virus core, Stanford University) were performed at about P42. Mice were anesthetized using isoflurane and mounted in stereotactic apparatus (Kopf). Coordinates for proximal CA1 were 1.4 mm lateral and 1.25 mm posterior from bregma, and 1.12 mm ventral from the brain surface. Virus was iontophoretically injected with current parameters 5 μA, 7 s on, 7 s off, for 2 min, using pipette tips with an outside perimeter of 10–15 μm. Mice were perfused about 2 weeks later and processed for immunostaining as described below.

## Immunostaining

Mice were injected with 2.5% Avertin and were transcardially perfused with PBS followed by 4% paraformaldehyde (PFA). Brains were dissected and post-fixed in 4% PFA overnight, and cryoprotected for about 24 hr in 30% sucrose. Brains were embedded in Optimum Cutting Temperature (OCT, Tissue-Tek), frozen in dry ice-cooled isopentane bath, and stored at –80 °C until sectioned. 60 μm thick floating sections were collected in PBS +0.02% sodium azide and stored at 4 °C. Sections were incubated in the following solutions at room temperature unless indicated: 1 hr in 0.3% PBS/Triton X-100 and 10% normal donkey serum, two nights in the primary antibody at 4 °C in 0.3% PBS/Triton X-100

and 10% normal donkey serum, 3 × 15 min in 0.3% PBS/Triton X-100, overnight in secondary antibody +DAPI (1:10,000 of 5 mg/ml, Sigma-Aldrich) in 0.3% PBS/Triton X-100 and 10% normal donkey serum, 2 × 15 min in 0.3% PBS/Triton X-100, and 15 min in PBS. Sections were mounted with Fluoromount-G (SouthernBiotech). Primary antibodies used were rat anti-mCherry (1:1000, ThermoFisher, M11217, RRID:AB_2536611), rabbit anti-Cre (1:500, Synaptic Systems, 257 003, RRID:AB_2619968), chicken anti-GFP (1:2500, Aves Labs, GFP-1020, RRID:AB_10000240) goat anti-FLAG (1:3000, Abcam, ab95045, RRID:AB_10676074) and rabbit anti-Lphn2 (1:500, Novus Biologicals, NBP2-58704). Secondary antibodies conjugated to Alexa 488, Alexa 568, or Cy3 (Jackson ImmunoResearch) were used at 1:500 from 50% glycerol stocks.

## Image and data analysis for CA1 axon tracing

Mice were only included if they passed the following criteria: (1) AAV injection site must be in proximal CA1 (most proximal 30%), (2) lentivirus injections sites must be in CA1 and not in the subiculum (*Figures 1, 3 and 4*) or lentivirus injections must be in the distal subiculum (*Figure 5*), (3) proximal CA1 axons must overlap with lentivirus injection site in the subiculum (*Figure 5*). All mice that fulfilled these criteria are reported in *Figures 1 and 3–5* and were included in quantifications. Images of injection sites (5x magnification) and projections (10x magnification) were acquired for every other 60 µm sagittal section using a Zeiss epifluorescence scope. Due to variations in injection sites within each mouse, exposure was adjusted for each mouse to avoid saturation. Fluorescence intensity measurements on unprocessed images were taken using FIJI and data processing was performed using MATLAB.

For injection site quantification, a 30-pixel-wide segmented line was drawn from proximal CA1 to distal CA1 using the DAPI signal as a guide. For projection quantification in the subiculum, a 200-pixel-wide segmented line was drawn from the proximal subiculum to the distal subiculum through the cell body layer using only DAPI as a guide. From this point, injection site and projection images were processed the same. Segmented lines were straightened using the 'Straighten' function, background subtraction was performed using the 'Subtract' function and intensity values were measured using the 'Plot Profile' command (FIJI). For injections that labeled both CA2 and proximal CA1, CA2 axons were present near the distal border of CA1 and spilled into the proximal subiculum. These axons had their intensity set to zero by using area selection and the clear function (FIJI). The intensity plots were resampled into 100 equal bins using a custom MATLAB code.

For trace quantification in *Figures 1, 3 and 4* the axon intensity was combined for all sections by summing all intensity values at each binned position. To calculate the fraction of axon intensity across proximal, mid, and distal subiculum the total axon intensity in bins 1–33, 34–66, and 67–100 were summed, respectively. The summed value from each of these regions was then divided by the total sum of axons from bins 1–100 to obtain the fraction of axons intensity within the proximal, mid, and distal subiculum. Fractions of axon intensities were compared using a two-way ANOVA with Sidak's multiple comparisons tests using Prism 9 (GraphPad). The mean position of the injection sites was calculated by generating a summed intensity trace as above and then multiplying the intensity value by the bin position, summing across the entire axis, and dividing by the sum of the intensity values. Representative images (*Figures 1, 3 and 4*) were taken using a Zeiss LSM 780 confocal microscope (20×magnification, tile scan, max projection).

In *Figure 5*, the experimental and data analysis procedures were identical to *Pederick et al., 2021* and, therefore, we used *LV-GFP* and *LV-GFP-P2A-Lphn2-FLAG* from that study to compare with *LV-GFP-P2A-Lphn2_F831A/M831A-FLAG* and *LV-GFP-P2A-Lphn2_T828G-FLAG* data generated from this study.

To quantify average axon intensity in GFP+ and adjacent GFP– regions in subiculum targets (*Figure 5F*), we restricted the analysis to the most distal 20% of the subiculum. To determine the GFP+ region we identified the intensity-weighted central row using the summed fluorescence of each row and determined the minimal symmetric window of rows around the central row that encompassed at least 50% of the total intensity in the restricted GFP image. This defined a rectangle in the original image that we designated as the GFP+ region. We then computed the mean fluorescence intensity in this region for the mCh channel. We used the two rows above and below (lateral and medial) the designated GFP+ region as the adjacent GFP– region and computed the mean mCh fluorescence across these four rows. To determine mCh fluorescence differences in GFP+ versus GFP– regions, we divided the mCh intensity in the GFP+ region by the mCh intensity in the GFP– region for each mouse

(i.e. GFP+/GFP–). mCh fluorescence intensity GFP+/GFP– was compared across groups using a one-way ANOVA with Tukey's multiple comparisons tests using Prism 9 (GraphPad). Three-dimensional mountain plots were generated using the 'surf' function.

## Quantification of overexpressed Lphn2 in CA1

P0 neonatal injections into CA1 and immunostaining were performed as stated above. 16-bit images were acquired with a Zeiss Axio Imager Z2 confocal microscope. All conditions were imaged with equal settings. For mean intensity quantification of FLAG and GFP a 50- and 20-pixel wide segmented line was drawn through the molecular layer and cell body layer of CA1, respectively, using the DAPI signal as a guide. The mean intensity of FLAG and GFP was calculated using the 'Measure' function.

From this point, only the section with the highest mean GFP expression was used from each animal. The FLAG mean intensity value was divided by the GFP mean intensity value to determine the FLAG/GFP mean intensity. FLAG/GFP mean intensities across conditions were compared using a one-way analysis of variance (ANOVA) with Tukey's multiple comparisons test in Prism 9 (GraphPad).

## Replicates

Technical replicates are defined here as repeated measurements of the same sample while biological replicates are defined as measurements of biologically distinct samples.

## Acknowledgements

We thank Z Li, T Li, C McLaughlin, D Wang, and Y Wu for critiques of the manuscript. We are also grateful to A Inoue (Tohoku University, Japan) for the generous gift of the HEKΔ7 cell line and Dr. Luke Lavis (Janelia Research Campus) for supplying the JF646 dye. This work was supported by NIH grants T32-MH015144 (NAPH), R01-NS050835 (LL), R01-MH54137 (JAJ), the Hope for Depression Research Foundation (JAJ), and P30EY012196 (ZH). LL is an investigator at the Howard Hughes Medical Institute.

## Additional information

### Funding

| Funder | Grant reference number | Author |
|---|---|---|
| National Institutes of Health | T32-MH015144 | Nicole A Perry-Hauser |
| National Institutes of Health | R01-NS050835 | Liqun Luo |
| National Institutes of Health | R01-MH54137 | Jonathan A Javitch |
| Hope for Depression Research Foundation | | Jonathan A Javitch |
| National Institutes of Health | P30EY012196 | Zhigang He |

The funders had no role in study design, data collection and interpretation, or the decision to submit the work for publication.

### Author contributions

Daniel T Pederick, Conceptualization, Resources, Data curation, Formal analysis, Validation, Investigation, Visualization, Writing – original draft, Writing – review and editing; Nicole A Perry-Hauser, Conceptualization, Resources, Data curation, Formal analysis, Funding acquisition, Validation, Investigation, Visualization, Writing – original draft, Writing – review and editing; Huyan Meng, Resources, Writing – review and editing; Zhigang He, Resources, Funding acquisition, Writing – review and editing; Jonathan A Javitch, Conceptualization, Resources, Data curation, Supervision, Funding acquisition, Writing – original draft, Writing – review and editing; Liqun Luo, Conceptualization, Resources,

Data curation, Supervision, Funding acquisition, Validation, Investigation, Visualization, Writing – original draft, Project administration, Writing – review and editing

### Author ORCIDs
Daniel T Pederick  http://orcid.org/0000-0003-1870-9475
Nicole A Perry-Hauser  http://orcid.org/0000-0003-3130-3023
Huyan Meng  http://orcid.org/0000-0003-1511-6156
Jonathan A Javitch  http://orcid.org/0000-0001-7395-2967
Liqun Luo  http://orcid.org/0000-0001-5467-9264

### Ethics
All procedures followed animal care and biosafety guidelines approved by Stanford University's Administrative Panel on Laboratory Animal Care (APLAC 14007) and Administrative Panel on Biosafety (APB-3669-LL120) in accordance with NIH guidelines.

### Decision letter and Author response
Decision letter https://doi.org/10.7554/eLife.83529.sa1
Author response https://doi.org/10.7554/eLife.83529.sa2

---

## Additional files

### Supplementary files
• MDAR checklist

### Data availability
All materials are available through requests to the corresponding authors. All custom code was identical to that reported in *Pederick et al., 2021* and can be accessed at https://github.com/dpederick/Reciprocal-repulsions-instruct-the-precise-assembly-of-parallel-hippocampal-networks/tree/1 (*Pederick, 2021*). All data generated or analyzed during this study are included in the manuscript and supporting file. Source data files have been provided for all figures.

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
