## [Editor Report]

This is an intriguing study investigating the molecular mechanisms of neural circuit developmental organization. Using a defined hippocampal circuit, the authors find that ectopic expression of an adhesion G protein receptor leads to axon mistargeting. This work defines new mechanisms of axon target specificity.

---

## [Decision Letter]

**Decision letter after peer review:**

Thank you for submitting your article "Context-dependent requirement of G protein coupling for Latrophilin-2 in target selection of hippocampal axons" for consideration by *eLife*. Your article has been reviewed by 3 peer reviewers, and the evaluation has been overseen by a Reviewing Editor and Marianne Bronner as the Senior Editor. The following individual involved in review of your submission has agreed to reveal their identity: Yimin Zou (Reviewer #3).

Essential revisions:

1) Examine ectopic Lphn2 expression, including the F831A/M835M mutant, and its impact on endogenous Ten3 expression.

2) Confirmation of Lphn2 localization in axons.

3) Clarify in the text that the presented results strongly suggest, but do not necessarily prove, that Lphn2 specifically couples with G-α 12/13 signaling in CA1, as this link has only been demonstrated in heterologous cells.

The authors can also consider the remaining reviewer comments, though these were not deemed to be essential.

*Reviewer #1 (Recommendations for the authors):*

– Figure 1—figure supplement 1: The F831A/M835A mutant expression looks qualitatively and quantitatively different from wt Lphn2 and T829G mutant (i.e., lower expression). If this is truly a representative image, it is problematic for interpreting these experiments since this is the mutant that fails to induce axon mistargeting. Could a more representative image be found? Or can the authors explain why this is not a problem?

– Figure 2B, 3B, 4A: The western blots in these figures contain many bands in addition to the expected cleaved and uncleaved Lphn2 protein. Do the authors have an explanation for this? Is this the result of Lphn2 proteolysis or aggregation? Can the western blot be cleaned up?

– Figure 5: Were all of the experiments in Figure 5 done at the same time? It seems that if any significant time passed between the experiments shown in 5B/C and 5D/E, then the latter pair are uncontrolled. (See legend to Figure 5. where the authors state that data from 5 B and C are from a previous publication).

– Figure 2-4. While these experiments convincingly show that Lphn2 can couple to Gα12/13 in heterologous cells, it is possible that they do not in neurons, or that they do in subiculum, but not (as expected) in CA1. As mentioned in the public review, the finding in this manuscript that Lphn2 signaling is context-dependent underscores the importance of this issue. While it is not reasonable to request a repeat of all the BRET experiments in neurons, much less in vivo, providing some type of evidence that Lphn2 couples to Gα12/13 in CA1 would strengthen this manuscript substantially. This could be done in a number of ways, including:

– Showing an interaction between Lphn2 and Gα12/13 (testing for other G-proteins) in neurons or in hippocampus via co-immunoprecipitation.

– Demonstrating an Lphn2-mediated increase in RhoA in neurons or hippocampus.

– Inhibiting RhoA and demonstrating that the ability of Lphn2 to redirect axons is compromised.

– Creating an Lphn2 mutant that does not interact with Gα12/13 and testing its ability to redirect axons.

– Creating an Lphn2 chimera that interacts with a different G-protein and tests its ability to redirect axons.

*Reviewer #2 (Recommendations for the authors):*

Specific recommendations related to presentation of data:

– Figure 1A – unclear what unlabeled red objects are meant to represent.

– Figure 2B/C -It's not clear to the specificity of Flag detection in the shown western blots. Could benefit from providing a negative control alongside Lphn2 transfected cells.

– Figure 2E – Graph is labelled "enterokinase-induced BRET", but enterokinase is only added to Lphn2 assay, correct? Signaling induction is relatively weak in comparison to their "control" Endothelin receptor, but this is activated by optimized ligand ET-1 induction. Based on this presentation, it is unclear as to the rationale for focus on Gq/G12/G13 signaling activation. Supplemental figure 2 better demonstrates G-protein selectivity, and as such would suggest remodeling main figure to include this data.

– Figure 3- From the quantifications, it appears that the mutant blocks GPCR signaling, but still has an impact on the axon targeting specificity. This suggests that there are GPCR signaling independent mechanisms that are also at work? This is briefly mentioned in the text, but could use additional discussion. Additionally, representative images of subiculum targeting for all conditions are not included, and should be provided.

– Figure 5 – Lack of representative images makes it difficult to assess quality and interpretability of experiment.

– Supplemental Figure 2S2 – Panel B mislabeled D1R? (labelled D2R in figure legend).

– Supplemental Figure 2/3/4 – Source data. Western blot lane descriptions not provided.

*Reviewer #3 (Recommendations for the authors):*

I have a suggestion for the authors to consider, probably by looking at the sections they already have, to see whether the misexpression of latrophilin-2 in the proximal CA1 may potentially change the expression of teneurin-3. If so, the interpretation of the results may be different or may need to include the possibility that the mistargeting of proximal axons to the proximal subiculum may be caused by a down regulation of teneurin-3.

---

## [Author Response]

Essential revisions:1) Examine ectopic Lphn2 expression, including the F831A/M835M mutant, and its impact on endogenous Ten3 expression.

This comment can be separated into two parts.

a) Is ectopic expression of Lphn2 mutants different to wild-type Lphn2?

We have performed additional experiments to quantify the levels of ectopic Lphn2 expression for wild type receptor and the two mutants. We found there to be no significant differences between expression levels after accounting for the injection efficiency.

Briefly, we injected *LV-GFP-P2A-Lphn2-FLAG*, *LV-GFP-P2A-Lphn2_F831A/M835A-FLAG* or *LVGFP-P2A-Lphn2_T829G-FLAG* into CA1 and quantified mean expression of GFP in CA1 cell body layer and mean expression of Lphn2, via FLAG, in the molecular layer of CA1. To account for injection variation, we calculated the amount of FLAG staining relative to GFP in the same section. We found no significant differences between Lphn2, Lphn2_F831A/M835A and Lphn2_T829G. Representative images and quantification can be found in the new Figure 1—figure supplement 2A-B.

In addition, we performed immunostaining with a Lphn2 antibody that detects both overexpressed and endogenous Lphn2. We found that the laser power required to visualize endogenous Lphn2 in the molecular layer of CA1 was ~19x of that required to observed overexpressed wild-type and mutant Lphn2. This suggests that even with variations of injection efficiency, the levels of overexpressed Lphn2 are likely substantially above that of endogenous Lphn2. Representative images can be found in the new Figure 1—figure supplement 2C-D.

b) Does ectopic expression of Lphn2 impact endogenous expression of Ten3?

We thank the reviewers for this comment and agree that it is an important question. We performed additional experiments to address this, but unfortunately technical limitations prevented us from reaching a clear-cut conclusion.

Ten3 immunostaining in CA1 is located within the molecular layer and is contributed both by Ten3 from proximal CA1 dendrites and from medial entorhinal cortex axons. This makes it extremely difficult to determine if changes in Ten3 protein are related to CA1 dendrites or invading axons. We previously showed that ectopic expression of Lphn2 causes repulsion of Ten3 axons, which would lead to a decrease of Ten3 immunostaining due to the absence of Ten3 positive axons. Therefore, by looking at Ten3 protein in the molecular layer we cannot determine if ectopic expression of Lphn2 impacts local expression of Ten3 or Ten3 expression from invading axons.

To overcome this issue, we aimed to analyze Ten3 expression in the cell body layer of CA1.

Unfortunately, we found that expression levels assayed by Ten3 antibody staining were very low in the cell body layer and comparable to background staining. This was also true even after performing Ten3 antibody adsorption on *Ten3* knockout brain sections.

Although we cannot rule out that Lphn2 ectopic expression cell-autonomously impacts Ten3 protein, we want to clarify that the proximal CA1 axon targeting phenotypes between Lphn2 OE and Ten3 KO conditions are different. When Ten3 is deleted from pCA1 axons in Ten3 conditional or whole animal knockout, pCA1 axons spread more proximally in subiculum while retaining reduced level in distal subiculum, compared to being restricted to distal subiculum in control animals. In contrast, when Lphn2 is ectopically expressed in pCA1 the axons completely shift and almost entirely target the most proximal part of subiculum.

The more severe mistargeting of pCA1 axons observed with Lphn2 overexpression compared to Ten3 KO suggests that the phenotype observed when Lphn2 is overexpressed in pCA1 axons is not due to impacting Ten3 expression alone. We have clarified this in the revised text (page 5).

2) Confirmation of Lphn2 localization in axons.

We have confirmed that FLAG staining is present in CA1 axons that target the subiculum in wild type Lphn2 and both mutant conditions. Representative images are now included in the new Figure 1— figure supplement 2A.

3) Clarify in the text that the presented results strongly suggest, but do not necessarily prove, that Lphn2 specifically couples with G-α 12/13 signaling in CA1, as this link has only been demonstrated in heterologous cells.

We have now emphasized in the text that these assays were performed in heterologous cells (see line 140, line 150, line 345) and state the following in the discussion:

“Thus, if Lphn2 coupling to Gα_12_/Gα_13_ also applies to CA1 neurons, which is strongly suggested by our results, Gα_12_/Gα_13_→RhoGEF→RhoA may be a plausible pathway for Lphn2 to mediate its function as a receptor for axon repulsion.” (line 350)

Reviewer #1 (Recommendations for the authors):– Figure 1—figure supplement 1: The F831A/M835A mutant expression looks qualitatively and quantitatively different from wt Lphn2 and T829G mutant (i.e., lower expression). If this is truly a representative image, it is problematic for interpreting these experiments since this is the mutant that fails to induce axon mistargeting. Could a more representative image be found? Or can the authors explain why this is not a problem?

We have performed additional experiments addressing this point and the new data are included. Please see our response in Essential Revision #1 above.

We have kept the previous data to confirm that *LV-Cre*, *LV-Cre-P2A-Lphn2-FLAG*, *LV-Cre-P2ALphn2_F831A/M835A-FLAG* and *LV-Cre-P2A-Lphn2_T829G-FLAG* are expressed when injected into CA1.

– Figure 2B, 3B, 4A: The western blots in these figures contain many bands in addition to the expected cleaved and uncleaved Lphn2 protein. Do the authors have an explanation for this? Is this the result of Lphn2 proteolysis or aggregation? Can the western blot be cleaned up?

To address Reviewer 1’s concerns about the quality of the immunoblots, we re-ran the experiment alongside an empty vector control. We also treated the samples for 1 hr with PNGase F (NEB, P0704S), an enzyme that removes *N*-linked oligosaccharides from proteins, to collapse bands with heterogeneous glycosylation and thereby “clean up” the blot. We observed that some of the additional bands in the gel were also present in the empty vector transfected cell line, suggesting that these were nonspecific interactions with our primary antibody. The new blots are found in Figure 2B, Figure 3B, Figure 4A, and Figure 2–4 – Source Data.

– Figure 5: Were all of the experiments in Figure 5 done at the same time? It seems that if any significant time passed between the experiments shown in 5B/C and 5D/E, then the latter pair are uncontrolled. (See legend to Figure 5. where the authors state that data from 5 B and C are from a previous publication).

The experiments in 5B/C and 5D/E were indeed performed at different times. We did not repeat the experiments shown in 5B/C as these experiments, involving matching two injections performed at different developmental stages, are very inefficient and take a large number of animals to complete. In addition, the data analysis is performed with internal normalization for each animal, where we assess the amount of mCh fluorescence in GFP-positive and GFP-negative areas for each animal.

All experimental and analysis steps were kept the same regardless of when the experiments were performed.

For the above reasons we thought it was unnecessary and unethical to repeat these control experiments in new animals.

– Figure 2-4. While these experiments convincingly show that Lphn2 can couple to Gα12/13 in heterologous cells, it is possible that they do not in neurons, or that they do in subiculum, but not (as expected) in CA1. As mentioned in the public review, the finding in this manuscript that Lphn2 signaling is context-dependent underscores the importance of this issue. While it is not reasonable to request a repeat of all the BRET experiments in neurons, much less in vivo, providing some type of evidence that Lphn2 couples to Gα12/13 in CA1 would strengthen this manuscript substantially.

We appreciate Reviewer 1’s concerns regarding the translation of our results from heterologous cells to neurons. We acknowledge that using heterologous cell lines comes with limitations, as there may be context related differences in signaling that emerge in neurons. For this revision, we have clarified that our in vitro findings are suggestive but do not prove this signaling also takes place in vivo (see our response to Essential Revision #3). We strongly agree that studying signaling in neurons and even better in slice or in vivo is important for the future. Unfortunately performing the additional experiments Reviewer 1 suggested is not straightforward, as we outlined below; however, we are working toward moving our research into a more native system for future efforts.

This could be done in a number of ways, including:– Showing an interaction between Lphn2 and Gα12/13 (testing for other G-proteins) in neurons or in hippocampus via co-immunoprecipitation.

Co-immunoprecipitation is unlikely to work for testing Lphn2 interaction with Gα12/13. This is because recruitment of G protein to Lphn2 would require activating the receptor and we currently do not have a way to activate these receptors in vivo. The interaction is also unlikely to be sufficiently stable to persist throughout the necessary purification steps.

– Demonstrating an Lphn2-mediated increase in RhoA in neurons or hippocampus.

There are several commercially-available kits for detecting RhoA activation following pull-down analysis. In brief, these kits use proteins that bind specifically to active RhoA to capture the protein for immunoblot analysis. While it is possible that we could test activation of RhoA in neuronal populations that have Lphn2 and corresponding mutants overexpressed, it will likely require extensive optimization to complete this series of experiments. Furthermore, again we have no way to activate in vivo, so we would only be assessing the impact of overexpression and not receptor activation.

– Inhibiting RhoA and demonstrating that the ability of Lphn2 to redirect axons is compromised.

This experimental design, while interesting, will take many animals to complete and is outside the scope of the current manuscript. Additionally, RhoA activation is downstream of many different signaling events and it will be difficult to attribute any biological effect directly to Lphn2 G protein signaling. We would also have to consider the experiment above to determine normal RhoA activity and likely RhoA expression prior to making any meaningful conclusions.

– Creating an Lphn2 mutant that does not interact with Gα12/13 and testing its ability to redirect axons.

While structural work has revealed coupling information for Gα12- and Gα13-bound Lphn3 (Qian et al., 2022), it is still unclear which mutations/residues selectively activate one pathway over the other. This is something that we are trying to achieve but is beyond the scope of the current study. It is possible that we could fully disrupt G-protein interaction by introducing T4 lysozyme into an intracellular loop or by truncating the cytoplasmic C-terminal sequences; however, this will likely also disrupt other signaling pathways (e.g., arrestin recruitment) and cannot be directly linked to one G protein (12/13) over another (q/s/o/i).

– Creating an Lphn2 chimera that interacts with a different G-protein and tests its ability to redirect axons.

As mentioned above, it remains unclear which mutations/residues selectively activate one pathway over another for Lphn2. Thus, this experiment is currently outside the scope of the submitted manuscript.

Reviewer #2 (Recommendations for the authors):Specific recommendations related to presentation of data:– Figure 1A – unclear what unlabeled red objects are meant to represent.

The red labels represent our previous finding of repulsive mechanisms that guide CA1 targeting in the subiculum. We have added additional text to the legend of Figure 1 to clarify this.

– Figure 2B/C -It's not clear to the specificity of Flag detection in the shown western blots. Could benefit from providing a negative control alongside Lphn2 transfected cells.

We have now included a negative control alongside the Lphn2-transfected cells. This can be found in Figure 2–4 – Source Data corresponding to the immunoblot. All immunoblots have been replaced with the re-run blot where the samples were treated for 1 hr with PNGase F to collapse bands with heterogenous glycosylation. The unspecified bands in the gel also present in the mock transfected lane likely result from nonspecific interactions with our primary antibody.

– Figure 2E – Graph is labelled "enterokinase-induced BRET", but enterokinase is only added to Lphn2 assay, correct? Signaling induction is relatively weak in comparison to their "control" Endothelin receptor, but this is activated by optimized ligand ET-1 induction. Based on this presentation, it is unclear as to the rationale for focus on Gq/G12/G13 signaling activation. Supplemental figure 2 better demonstrates G-protein selectivity, and as such would suggest remodeling main figure to include this data.

We thank Reviewer 2 for noticing the error in the y-axis of Figure 2E. We have now changed the label to ‘ligand-induced BRET.” We have also added the data for the remaining G proteins to better demonstrate selectivity as suggested. We also want to note that the purpose of the controls was to demonstrate that our experimental setup was working, not to compare signaling induction across receptors. To emphasize that the focus should be on Lphn2 activation, we modified the y-axis to better reflect the Lphn2 data, which clearly shows that G12 and G13 are far more actively activated compared to the other G proteins.

– Figure 3- From the quantifications, it appears that the mutant blocks GPCR signaling, but still has an impact on the axon targeting specificity. This suggests that there are GPCR signaling independent mechanisms that are also at work? This is briefly mentioned in the text, but could use additional discussion. Additionally, representative images of subiculum targeting for all conditions are not included, and should be provided.

We thank Reviewer 2 for highlighting the difference between F831A/M835A mutant and control in Figure 3E. In light of this, we have added the following to the Discussion:

“However, given the statistically significant difference in total axon intensity in dSub between control and Lphn2_F831A/M835A (Figure 3E), we cannot rule out some residual tethered agonist-independent, G protein-mediated signaling or a Lphn2 function independent of G protein signaling, at least in the context of overexpression.”

The LV-Cre and LV-Cre_Lphn2 conditions are presented in Figure 1. We did not show them again in Figure 3 to avoid duplication.

– Figure 5 – Lack of representative images makes it difficult to assess quality and interpretability of experiment.

Representative images of pCA1 axons targeting GFP-ve and GFP+ve sections for each condition are included in a new Figure 5—figure supplement 2.

– Supplemental Figure 2S2 – Panel B mislabeled D1R? (labelled D2R in figure legend).

We thank Reviewer 2 for pointing out this discrepancy. The Panel B of Supplemental Figure 2 is now correctly labeled D2R.

– Supplemental Figure 2/3/4 – Source data. Western blot lane descriptions not provided.

We have now included the lane descriptions for the immunoblot source data.

Reviewer #3 (Recommendations for the authors):I have a suggestion for the authors to consider, probably by looking at the sections they already have, to see whether the misexpression of latrophilin-2 in the proximal CA1 may potentially change the expression of teneurin-3. If so, the interpretation of the results may be different or may need to include the possibility that the mistargeting of proximal axons to the proximal subiculum may be caused by a down regulation of teneurin-3.

Thank you. Please see our response in “Essential Revision #1” above.